# IMPROVED TECHNIQUES FOR TRAINING CONSISTENCY MODELS

**Yang Song & Prafulla Dhariwal**
OpenAI
{songyang,prafulla}@openai.com

## ABSTRACT

Consistency models are a nascent family of generative models that can sample high quality data in one step without the need for adversarial training. Current consistency models achieve optimal sample quality by distilling from pre-trained diffusion models and employing learned metrics such as LPIPS. However, distillation limits the quality of consistency models to that of the pre-trained diffusion model, and LPIPS causes undesirable bias in evaluation. To tackle these challenges, we present improved techniques for *consistency training*, where consistency models learn directly from data without distillation. We delve into the theory behind consistency training and identify a previously overlooked flaw, which we address by eliminating Exponential Moving Average from the teacher consistency model. To replace learned metrics like LPIPS, we adopt Pseudo-Huber losses from robust statistics. Additionally, we introduce a lognormal noise schedule for the consistency training objective, and propose to double total discretization steps every set number of training iterations. Combined with better hyperparameter tuning, these modifications enable consistency models to achieve FID scores of 2.51 and 3.25 on CIFAR-10 and ImageNet $64 \times 64$ respectively in a single sampling step. These scores mark a $3.5\times$ and $4\times$ improvement compared to prior consistency training approaches. Through two-step sampling, we further reduce FID scores to 2.24 and 2.77 on these two datasets, surpassing those obtained via distillation in both one-step and two-step settings, while narrowing the gap between consistency models and other state-of-the-art generative models.

## 1 INTRODUCTION

Consistency models (Song et al., 2023) are an emerging family of generative models that produce high-quality samples using a single network evaluation. Unlike GANs (Goodfellow et al., 2014), consistency models are not trained with adversarial optimization and thus sidestep the associated training difficulty. Compared to score-based diffusion models (Sohl-Dickstein et al., 2015; Song & Ermon, 2019; 2020; Ho et al., 2020; Song et al., 2021), consistency models do not require numerous sampling steps to generate high-quality samples. They are trained to generate samples in a single step, but still retain important advantages of diffusion models, such as the flexibility to exchange compute for sample quality through multistep sampling, and the ability to perform zero-shot data editing.

We can train consistency models using either consistency distillation (CD) or consistency training (CT). The former requires pre-training a diffusion model and distilling the knowledge therein into a consistency model. The latter allows us to train consistency models directly from data, establishing them as an independent family of generative models. Previous work (Song et al., 2023) demonstrates that CD significantly outperforms CT. However, CD adds computational overhead to the training process since it requires learning a separate diffusion model. Additionally, distillation limits the sample quality of the consistency model to that of the diffusion model. To avoid the downsides of CD and to position consistency models as an independent family of generative models, we aim to improve CT to either match or exceed the performance of CD.

For optimal sample quality, both CD and CT rely on learned metrics like the Learned Perceptual Image Patch Similarity (LPIPS) (Zhang et al., 2018) in previous work (Song et al., 2023). However, depending on LPIPS has two primary downsides. Firstly, there could be potential bias in evaluation

since the same ImageNet dataset (Deng et al., 2009) trains both LPIPS and the Inception network in Fréchet Inception Distance (FID) (Heusel et al., 2017), which is the predominant metric for image quality. As analyzed in Kynkäänniemi et al. (2023), improvements of FIDs can come from accidental leakage of ImageNet features from LPIPS, causing inflated FID scores. Secondly, learned metrics require pre-training auxiliary networks for feature extraction. Training with these metrics requires backpropagating through extra neural networks, which increases the demand for compute.

To tackle these challenges, we introduce improved techniques for CT that not only surpass CD in sample quality but also eliminate the dependence on learned metrics like LPIPS. Our techniques are motivated from both theoretical analysis, and comprehensive experiments on the CIFAR-10 dataset (Krizhevsky et al., 2014). Specifically, we perform an in-depth study on the empirical impact of weighting functions, noise embeddings, and dropout in CT. Additionally, we identify an overlooked flaw in prior theoretical analysis for CT and propose a simple fix by removing the Exponential Moving Average (EMA) from the teacher network. We adopt Pseudo-Huber losses from robust statistics to replace LPIPS. Furthermore, we study how sample quality improves as the number of discretization steps increases, and utilize the insights to propose a simple but effective curriculum for total discretization steps. Finally, we propose a new schedule for sampling noise levels in the CT objective based on lognormal distributions.

Taken together, these techniques allow CT to attain FID scores of 2.51 and 3.25 for CIFAR-10 and ImageNet $64 \times 64$ in one sampling step, respectively. These scores not only surpass CD but also represent improvements of $3.5\times$ and $4\times$ over previous CT methods. Furthermore, they significantly outperform the best few-step diffusion distillation techniques for diffusion models even without the need for distillation. By two-step generation, we achieve improved FID scores of 2.24 and 2.77 on CIFAR-10 and ImageNet $64 \times 64$, surpassing the scores from CD in both one-step and two-step settings. These results rival many top-tier diffusion models and GANs, showcasing the strong promise of consistency models as a new independent family of generative models.

## 2 CONSISTENCY MODELS

Central to the formulation of consistency models is the probability flow ordinary differential equation (ODE) from Song et al. (2021). Let us denote the data distribution by $p_{\text{data}}(\mathbf{x})$. When we add Gaussian noise with mean zero and standard deviation $\sigma$ to this data, the resulting perturbed distribution is given by $p_\sigma(\mathbf{x}) = \int p_{\text{data}}(\mathbf{y}) \mathcal{N}(\mathbf{x} \mid \mathbf{y}, \sigma^2 \boldsymbol{I}) \, \mathrm{d}\mathbf{y}$. The probability flow ODE, as presented in Karras et al. (2022), takes the form of

$$\frac{\mathrm{d}\mathbf{x}}{\mathrm{d}\sigma} = -\sigma \nabla_{\mathbf{x}} \log p_\sigma(\mathbf{x}) \quad \sigma \in [\sigma_{\min}, \sigma_{\max}], \tag{1}$$

where the term $\nabla_{\mathbf{x}} \log p_\sigma(\mathbf{x})$ is known as the *score function* of $p_\sigma(\mathbf{x})$ (Song et al., 2019; Song & Ermon, 2019; 2020; Song et al., 2021). Here $\sigma_{\min}$ is a small positive value such that $p_{\sigma_{\min}}(\mathbf{x}) \approx p_{\text{data}}(\mathbf{x})$, introduced to avoid numerical issues in ODE solving. Meanwhile, $\sigma_{\max}$ is sufficiently large so that $p_\sigma(\mathbf{x}) \approx \mathcal{N}(\mathbf{0}, \sigma_{\max}^2 \mathbf{I})$. Following Karras et al. (2022); Song et al. (2023), we adopt $\sigma_{\min} = 0.002$, and $\sigma_{\max} = 80$ throughout the paper. Crucially, solving the probability flow ODE from noise level $\sigma_1$ to $\sigma_2$ allows us to transform a sample $\mathbf{x}_{\sigma_1} \sim p_{\sigma_1}(\mathbf{x})$ into $\mathbf{x}_{\sigma_2} \sim p_{\sigma_2}(\mathbf{x})$.

The ODE in Eq. (1) establishes a bijective mapping between a noisy data sample $\mathbf{x}_\sigma \sim p_\sigma(\mathbf{x})$ and $\mathbf{x}_{\sigma_{\min}} \sim p_{\sigma_{\min}}(\mathbf{x}) \approx p_{\text{data}}(\mathbf{x})$. This mapping, denoted as $\boldsymbol{f}^* : (\mathbf{x}_\sigma, \sigma) \mapsto \mathbf{x}_{\sigma_{\min}}$, is termed the *consistency function*. By its very definition, the consistency function satisfies the *boundary condition* $\boldsymbol{f}^*(\mathbf{x}, \sigma_{\min}) = \mathbf{x}$. A *consistency model*, which we denote by $\boldsymbol{f}_{\boldsymbol{\theta}}(\mathbf{x}, \sigma)$, is a neural network trained to approximate the consistency function $\boldsymbol{f}^*(\mathbf{x}, \sigma)$. To meet the boundary condition, we follow Song et al. (2023) to parameterize the consistency model as

$$\boldsymbol{f}_{\boldsymbol{\theta}}(\mathbf{x}, \sigma) = c_{\text{skip}}(\sigma)\mathbf{x} + c_{\text{out}}(\sigma) \boldsymbol{F}_{\boldsymbol{\theta}}(\mathbf{x}, \sigma), \tag{2}$$

where $\boldsymbol{F}_{\boldsymbol{\theta}}(\mathbf{x}, \sigma)$ is a free-form neural network, while $c_{\text{skip}}(\sigma)$ and $c_{\text{out}}(\sigma)$ are differentiable functions such that $c_{\text{skip}}(\sigma_{\min}) = 1$ and $c_{\text{out}}(\sigma_{\min}) = 0$.

To train the consistency model, we discretize the probability flow ODE using a sequence of noise levels $\sigma_{\min} = \sigma_1 < \sigma_2 < \cdots < \sigma_N = \sigma_{\max}$, where we follow Karras et al. (2022); Song et al. (2023) in setting $\sigma_i = (\sigma_{\min}^{1/\rho} + \frac{i-1}{N-1}(\sigma_{\max}^{1/\rho} - \sigma_{\min}^{1/\rho}))^\rho$ for $i \in [\![1, N]\!]$, and $\rho = 7$, where $[\![a, b]\!]$ denotes the set

of integers $\{a, a+1, \cdots, b\}$. The model is trained by minimizing the following *consistency matching* (CM) loss over $\boldsymbol{\theta}$:

$$\mathcal{L}^N(\boldsymbol{\theta}, \boldsymbol{\theta}^-) = \mathbb{E}\left[\lambda(\sigma_i) d(\boldsymbol{f_\theta}(\mathbf{x}_{\sigma_{i+1}}, \sigma_{i+1}), \boldsymbol{f_{\theta^-}}(\breve{\mathbf{x}}_{\sigma_i}, \sigma_i))\right], \qquad (3)$$

where $\breve{\mathbf{x}}_{\sigma_i} = \mathbf{x}_{\sigma_{i+1}} - (\sigma_i - \sigma_{i+1})\sigma_{i+1}\nabla_\mathbf{x} \log p_{\sigma_{i+1}}(\mathbf{x})|_{\mathbf{x}=\mathbf{x}_{\sigma_{i+1}}}$. In Eq. (3), $d(\boldsymbol{x}, \boldsymbol{y})$ is a metric function comparing vectors $\boldsymbol{x}$ and $\boldsymbol{y}$, and $\lambda(\sigma) > 0$ is a weighting function. Typical metric functions include the squared $\ell_2$ metric $d(\boldsymbol{x}, \boldsymbol{y}) = \|\boldsymbol{x} - \boldsymbol{y}\|_2^2$, and the Learned Perceptual Image Patch Similarity (LPIPS) metric introduced in Zhang et al. (2018). The expectation in Eq. (3) is taken over the following sampling process: $i \sim \mathcal{U}[\![1, N-1]\!]$ where $\mathcal{U}[\![1, N-1]\!]$ represents the uniform distribution over $\{1, 2, \cdots, N-1\}$, and $\mathbf{x}_{\sigma_{i+1}} \sim p_{\sigma_{i+1}}(\mathbf{x})$. Note that $\breve{\mathbf{x}}_{\sigma_i}$ is derived from $\mathbf{x}_{\sigma_{i+1}}$ by solving the probability flow ODE in the reverse direction for a single step. In Eq. (3), $\boldsymbol{f_\theta}$ and $\boldsymbol{f_{\theta^-}}$ are referred to as the *student network* and the *teacher network*, respectively. The teacher's parameter $\boldsymbol{\theta}^-$ is obtained by applying Exponential Moving Average (EMA) to the student's parameter $\boldsymbol{\theta}$ during the course of training as follows:

$$\boldsymbol{\theta}^- \leftarrow \mathrm{stopgrad}(\mu\boldsymbol{\theta}^- + (1-\mu)\boldsymbol{\theta}), \qquad (4)$$

with $0 \leqslant \mu < 1$ representing the EMA decay rate. Here we explicitly employ the $\mathrm{stopgrad}$ operator to highlight that the teacher network remains fixed for each optimization step of the student network. However, in subsequent discussions, we will omit the $\mathrm{stopgrad}$ operator when its presence is clear and unambiguous. In practice, we also maintain EMA parameters for the student network to achieve better sample quality at inference time. It is clear that as $N$ increases, the consistency model optimized using Eq. (3) approaches the true consistency function. For faster training, Song et al. (2023) propose a curriculum learning strategy where $N$ is progressively increased and the EMA decay rate $\mu$ is adjusted accordingly. This curriculum for $N$ and $\mu$ is denoted by $N(k)$ and $\mu(k)$, where $k \in \mathbb{N}$ is a non-negative integer indicating the current training step.

Given that $\breve{\mathbf{x}}_{\sigma_i}$ relies on the unknown score function $\nabla_\mathbf{x} \log p_{\sigma_{i+1}}(\mathbf{x})$, directly optimizing the consistency matching objective in Eq. (3) is infeasible. To circumvent this challenge, Song et al. (2023) propose two training algorithms: *consistency distillation* (CD) and *consistency training* (CT). For consistency distillation, we first train a diffusion model $\boldsymbol{s_\phi}(\mathbf{x}, \sigma)$ to estimate $\nabla_\mathbf{x} \log p_\sigma(\mathbf{x})$ via score matching (Hyvärinen, 2005; Vincent, 2011; Song et al., 2019; Song & Ermon, 2019), then approximate $\breve{\mathbf{x}}_{\sigma_i}$ with $\hat{\mathbf{x}}_{\sigma_i} = \mathbf{x}_{\sigma_{i+1}} - (\sigma_i - \sigma_{i+1})\sigma_{i+1}\boldsymbol{s_\phi}(\mathbf{x}_{\sigma_{i+1}}, \sigma_{i+1})$. On the other hand, consistency training employs a different approximation method. Recall that $\mathbf{x}_{\sigma_{i+1}} = \mathbf{x} + \sigma_{i+1}\mathbf{z}$ with $\mathbf{x} \sim p_{\mathrm{data}}(\mathbf{x})$ and $\mathbf{z} \sim \mathcal{N}(\mathbf{0}, \boldsymbol{I})$. Using the same x and z, Song et al. (2023) define $\breve{\mathbf{x}}_{\sigma_i} = \mathbf{x} + \sigma_i\mathbf{z}$ as an approximation to $\breve{\mathbf{x}}_{\sigma_i}$, which leads to the consistency training objective below:

$$\mathcal{L}_{\mathrm{CT}}^N(\boldsymbol{\theta}, \boldsymbol{\theta}^-) = \mathbb{E}\left[\lambda(\sigma_i) d(\boldsymbol{f_\theta}(\mathbf{x} + \sigma_{i+1}\mathbf{z}, \sigma_{i+1}), \boldsymbol{f_{\theta^-}}(\mathbf{x} + \sigma_i\mathbf{z}, \sigma_i))\right]. \qquad (5)$$

As analyzed in Song et al. (2023), this objective is asymptotically equivalent to consistency matching in the limit of $N \to \infty$. We will revisit this analysis in Section 3.2.

After training a consistency model $\boldsymbol{f_\theta}(\mathbf{x}, \sigma)$ through CD or CT, we can directly generate a sample x by starting with $\mathbf{z} \sim \mathcal{N}(\mathbf{0}, \sigma_{\max}^2\boldsymbol{I})$ and computing $\mathbf{x} = \boldsymbol{f_\theta}(\mathbf{z}, \sigma_{\max})$. Notably, these models also enable multistep generation. For a sequence of indices $1 = i_1 < i_2 < \cdots < i_K = N$, we start by sampling $\mathbf{x}_K \sim \mathcal{N}(\mathbf{0}, \sigma_{\max}^2\boldsymbol{I})$ and then iteratively compute $\mathbf{x}_k \leftarrow \boldsymbol{f_\theta}(\mathbf{x}_{k+1}, \sigma_{i_{k+1}}) + \sqrt{\sigma_{i_k}^2 - \sigma_{\min}^2}\mathbf{z}_k$ for $k = K-1, K-2, \cdots, 1$, where $\mathbf{z}_k \sim \mathcal{N}(\mathbf{0}, \boldsymbol{I})$. The resulting sample $\mathbf{x}_1$ approximates the distribution $p_{\mathrm{data}}(\mathbf{x})$. In our experiments, setting $K = 3$ (two-step generation) often enhances the quality of one-step generation considerably, though increasing the number of sampling steps further provides diminishing benefits.

# 3 IMPROVED TECHNIQUES FOR CONSISTENCY TRAINING

Below we re-examine the design choices of CT in Song et al. (2023) and pinpoint modifications that improve its performance, which we summarize in Table 1. We focus on CT without learned metric functions. For our experiments, we employ the Score SDE architecture in Song et al. (2021) and train the consistency models for 400,000 iterations on the CIFAR-10 dataset (Krizhevsky et al., 2014) without class labels. While our primary focus remains on CIFAR-10 in this section, we observe similar improvements on other datasets, including ImageNet $64 \times 64$ (Deng et al., 2009). We measure sample quality using Fréchet Inception Distance (FID) (Heusel et al., 2017).

Table 1: Comparing the design choices for CT in Song et al. (2023) versus our modifications.

| | Design choice in Song et al. (2023) | Our modifications |
|---|---|---|
| EMA decay rate for the teacher network | $\mu(k) = \exp\left(\frac{s_0 \log \mu_0}{N(k)}\right)$ | $\mu(k) = 0$ |
| Metric in consistency loss | $d(\boldsymbol{x}, \boldsymbol{y}) = \text{LPIPS}(\boldsymbol{x}, \boldsymbol{y})$ | $d(\boldsymbol{x}, \boldsymbol{y}) = \sqrt{\|\boldsymbol{x} - \boldsymbol{y}\|_2^2 + c^2} - c$ |
| Discretization curriculum | $N(k) = \left\lceil \sqrt{\frac{k}{K}((s_1 + 1)^2 - s_0^2) + s_0^2} - 1 \right\rceil + 1$ | $N(k) = \min(s_0 2^{\lfloor \frac{k}{K'} \rfloor}, s_1) + 1,$ 
 where $K' = \left\lfloor \frac{K}{\log_2\lfloor s_1/s_0 \rfloor + 1} \right\rfloor$ |
| Noise schedule | $\sigma_i$, where $i \sim \mathcal{U}[\![1, N(k) - 1]\!]$ | $\sigma_i$, where $i \sim p(i)$, and $p(i) \propto$ 
 $\text{erf}\left(\frac{\log(\sigma_{i+1}) - P_{\text{mean}}}{\sqrt{2} P_{\text{std}}}\right) - \text{erf}\left(\frac{\log(\sigma_i) - P_{\text{mean}}}{\sqrt{2} P_{\text{std}}}\right)$ |
| Weighting function | $\lambda(\sigma_i) = 1$ | $\lambda(\sigma_i) = \frac{1}{\sigma_{i+1} - \sigma_i}$ |
| Parameters | $s_0 = 2, s_1 = 150, \mu_0 = 0.9$ on CIFAR-10 

 $s_0 = 2, s_1 = 200, \mu_0 = 0.95$ on ImageNet $64 \times 64$ | $s_0 = 10, s_1 = 1280$ 
 $c = 0.00054\sqrt{d}$, $d$ is data dimensionality 
 $P_{\text{mean}} = -1.1, P_{\text{std}} = 2.0$ |
| | $k \in [\![0, K]\!]$, where $K$ is the total training iterations | |
| | $\sigma_i = (\sigma_{\min}^{1/\rho} + \frac{i-1}{N(k)-1}(\sigma_{\max}^{1/\rho} - \sigma_{\min}^{1/\rho}))^\rho$, where $i \in [\![1, N(k)]\!], \rho = 7, \sigma_{\min} = 0.002, \sigma_{\max} = 80$ | |

## 3.1 WEIGHTING FUNCTIONS, NOISE EMBEDDINGS, AND DROPOUT

We start by exploring several hyperparameters that are known to be important for diffusion models, including the weighting function $\lambda(\sigma)$, the embedding layer for noise levels, and dropout (Ho et al., 2020; Song et al., 2021; Dhariwal & Nichol, 2021; Karras et al., 2022). We find that proper selection of these hyperparameters greatly improve CT when using the squared $\ell_2$ metric.

The default weighting function in Song et al. (2023) is uniform, *i.e.*, $\lambda(\sigma) \equiv 1$. This assigns equal weights to consistency losses at all noise levels, which we find to be suboptimal. We propose to modify the weighting function so that it reduces as noise levels increase. The rationale is that errors from minimizing consistency losses in smaller noise levels can influence larger ones and therefore should be weighted more heavily. Specifically, our weighting function (*cf*., Table 1) is defined as $\lambda(\sigma_i) = \frac{1}{\sigma_{i+1} - \sigma_i}$. The default choice for $\sigma_i$, given in Section 2, ensures that $\lambda(\sigma_i) = \frac{1}{\sigma_{i+1} - \sigma_i}$ reduces monotonically as $\sigma_i$ increases, thus assigning smaller weights to higher noise levels. As shown in Fig. 1c, this refined weighting function notably improves the sample quality in CT with the squared $\ell_2$ metric.

In Song et al. (2023), Fourier embedding layers (Tancik et al., 2020) and positional embedding layers (Vaswani et al., 2017) are used to embed noise levels for CIFAR-10 and ImageNet $64 \times 64$ respectively. It is essential that noise embeddings are sufficiently sensitive to minute differences to offer training signals, yet too much sensitivity can lead to training instability. As shown in Fig. 1b, high sensitivity can lead to the divergence of continuous-time CT (Song et al., 2023). This is a known challenge in Song et al. (2023), which they circumvent by initializing the consistency model with parameters from a pre-trained diffusion model. In Fig. 1b, we show continuous-time CT on CIFAR-10 converges with random initial parameters, provided we use a less sensitive noise embedding layer with a reduced Fourier scale parameter, as visualized in Fig. 1a. For discrete-time CT, models are less affected by the sensitivity of the noise embedding layers, but as shown in Fig. 1c, reducing the scale parameter in Fourier embedding layers from the default value of 16.0 to a smaller value of 0.02 still leads to slight improvement of FIDs on CIFAR-10. For ImageNet models, we employ the default positional embedding, as it has similar sensitivity to Fourier embedding with scale 0.02 (see Fig. 1a).

Previous experiments with consistency models in Song et al. (2023) always employ zero dropout, motivated by the fact that consistency models generate samples in a single step, unlike diffusion models that do so in multiple steps. Therefore, it is intuitive that consistency models, facing a more challenging task, would be less prone to overfitting and need less regularization than their diffusion counterparts. Contrary to our expectations, we discovered that using larger dropout than diffusion models improves the sample quality of consistency models. Specifically, as shown in Fig. 1c, a dropout rate of 0.3 for consistency models on CIFAR-10 obtains better FID scores. For ImageNet $64 \times 64$, we find it beneficial to apply dropout of 0.2 to layers with resolution less than or equal to $16 \times 16$, following Hoogeboom et al. (2023). We additionally ensure that the random number

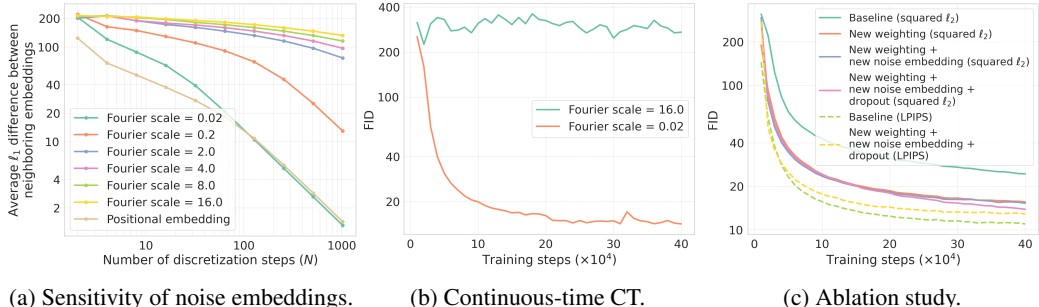

(a) Sensitivity of noise embeddings.    (b) Continuous-time CT.    (c) Ablation study.

Figure 1: (a) As the Fourier scale parameter decreases, Fourier noise embeddings become less sensitive to minute noise differences. This sensitivity is closest to that of positional embeddings when the Fourier scale is set to 0.02. (b) Continuous-time CT diverges when noise embeddings are overly sensitive to minor noise differences. (c) An ablation study examines the effects of our selections for weighting function ($\frac{1}{\sigma_{i+1}-\sigma_i}$), noise embedding (Fourier scale = 0.02), and dropout (= 0.3) on CT using the squared $\ell_2$ metric. Here baseline models for both metrics follow configurations in Song et al. (2023). All models are trained on CIFAR-10 without class labels.

generators for dropout share the same states across the student and teacher networks when optimizing the CT objective in Eq. (5).

By choosing the appropriate weighting function, noise embedding layers, and dropout, we significantly improve the sample quality of consistency models using the squared $\ell_2$ metric, closing the gap with the original CT in Song et al. (2023) that relies on LPIPS (see Fig. 1c). Although our modifications do not immediately improve the sample quality of CT with LPIPS, combining with additional techniques in Section 3.2 will yield significant improvements for both metrics.

### 3.2 REMOVING EMA FOR THE TEACHER NETWORK

When training consistency models, we minimize the discrepancy between models evaluated at adjacent noise levels. Recall from Section 2 that the model with the lower noise level is termed the *teacher network*, and its counterpart the *student network*. While Song et al. (2023) maintains EMA parameters for both networks with potentially varying decay rates, we present a theoretical argument indicating that the EMA decay rate for the teacher network should always be zero for CT, although it can be nonzero for CD. We revisit the theoretical analysis in Song et al. (2023) to support our assertion and provide empirical evidence that omitting EMA from the teacher network in CT notably improves the sample quality of consistency models.

To support the use of CT, Song et al. (2023) present two theoretical arguments linking the CT and CM objectives as $N \to \infty$. The first line of reasoning, which we call Argument (i), draws upon Theorem 2 from Song et al. (2023) to show that under certain regularity conditions, $\mathcal{L}_{\text{CT}}^N(\boldsymbol{\theta}, \boldsymbol{\theta}^-) = \mathcal{L}^N(\boldsymbol{\theta}, \boldsymbol{\theta}^-) + o(\Delta\sigma)$. That is, when $N \to \infty$, we have $\Delta\sigma \to 0$ and hence $\mathcal{L}_{\text{CT}}^N(\boldsymbol{\theta}, \boldsymbol{\theta}^-)$ converges to $\mathcal{L}^N(\boldsymbol{\theta}, \boldsymbol{\theta}^-)$ asymptotically. The second argument, called Argument (ii), is grounded in Theorem 6 from Song et al. (2023) which asserts that when $\boldsymbol{\theta}^- = \boldsymbol{\theta}$, both $\lim_{N\to\infty}(N-1)\nabla_{\boldsymbol{\theta}}\mathcal{L}^N(\boldsymbol{\theta}, \boldsymbol{\theta}^-)$ and $\lim_{N\to\infty}(N-1)\nabla_{\boldsymbol{\theta}}\mathcal{L}_{\text{CT}}^N(\boldsymbol{\theta}, \boldsymbol{\theta}^-)$ are well-defined and identical. This suggests that after scaling by $N-1$, gradients of the CT and CM objectives match in the limit of $N \to \infty$, leading to equivalent training dynamics. Unlike Argument (i), Argument (ii) is valid only when $\boldsymbol{\theta}^- = \boldsymbol{\theta}$, which can be enforced by setting the EMA decay rate $\mu$ for the teacher network to zero in Eq. (4).

We show this inconsistency in requirements for Argument (i) and (ii) to hold is caused by flawed theoretical analysis of the former. Specifically, Argument (i) fails if $\lim_{N\to\infty}\mathcal{L}^N(\boldsymbol{\theta}, \boldsymbol{\theta}^-)$ is not a valid objective for learning consistency models, which we show can happen when $\boldsymbol{\theta}^- \neq \boldsymbol{\theta}$. To give a concrete example, consider a data distribution $p_{\text{data}}(x) = \delta(x - \xi)$, which leads to $p_\sigma(x) = \mathcal{N}(x; \xi, \sigma^2)$ and a ground truth consistency function $f^*(x, \sigma) = \frac{\sigma_{\min}}{\sigma}x + \left(1 - \frac{\sigma_{\min}}{\sigma}\right)\xi$. Let us define the consistency model as $f_\theta(x, \sigma) = \frac{\sigma_{\min}}{\sigma}x + \left(1 - \frac{\sigma_{\min}}{\sigma}\right)\theta$. In addition, let $\sigma_i = \sigma_{\min} + \frac{i-1}{N-1}(\sigma_{\max} - \sigma_{\min})$ for $i \in [\![1, N]\!]$ be the noise levels, where we have $\Delta\sigma = \frac{\sigma_{\max}-\sigma_{\min}}{N-1}$. Given $z \sim \mathcal{N}(0, 1)$ and $x_{\sigma_{i+1}} = \xi + \sigma_{i+1}z$, it is straightforward to show that $\breve{x}_{\sigma_i} = x_{\sigma_{i+1}} - \sigma_{i+1}(\sigma_i - \sigma_{i+1})\nabla_x \log p_\sigma(x_{\sigma_{i+1}})$

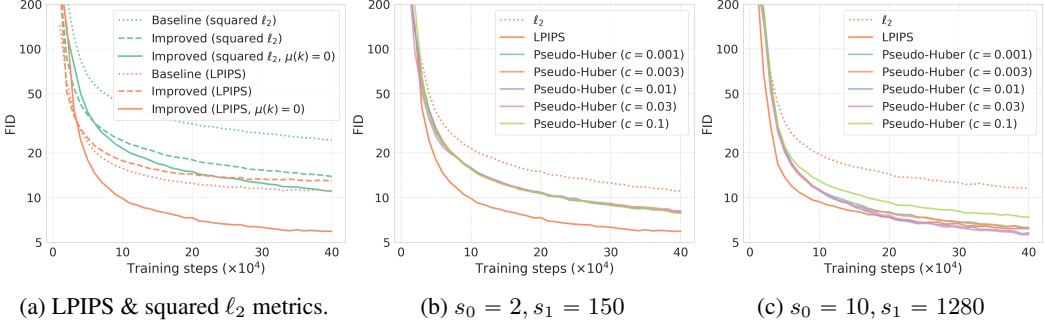

Figure 2: (a) Removing EMA in the teacher network leads to significant improvement in FIDs. (b, c) Pseudo-Huber metrics significantly improve the sample quality of squared $\ell_2$ metric, and catches up with LPIPS when using overall larger $N(k)$, where the Pseudo-Huber metric with $c = 0.03$ is the optimal. All training runs here employ the improved techniques from Sections 3.1 and 3.2.

simplifies to $\tilde{x}_{\sigma_i} = \xi + \sigma_i z$. As a result, the objectives for CM and CT align perfectly in this toy example. Building on top of this analysis, the following result proves that $\lim_{N \to \infty} \mathcal{L}^N(\theta, \theta^-)$ here is not amenable for learning consistency models whenever $\theta^- \neq \theta$.

**Proposition 1.** *Given the notations introduced earlier, and using the uniform weighting function $\lambda(\sigma) = 1$ along with the squared $\ell_2$ metric, we have*

$$\lim_{N \to \infty} \mathcal{L}^N(\theta, \theta^-) = \lim_{N \to \infty} \mathcal{L}_{CT}^N(\theta, \theta^-) = \mathbb{E}\left[\left(1 - \frac{\sigma_{min}}{\sigma_i}\right)^2 (\theta - \theta^-)^2\right] \quad \text{if } \theta^- \neq \theta \tag{6}$$

$$\lim_{N \to \infty} \frac{1}{\Delta\sigma} \frac{\mathrm{d}\mathcal{L}^N(\theta, \theta^-)}{\mathrm{d}\theta} = \begin{cases} \frac{\mathrm{d}}{\mathrm{d}\theta}\mathbb{E}\left[\frac{\sigma_{min}}{\sigma_i^2}\left(1 - \frac{\sigma_{min}}{\sigma_i}\right)(\theta - \xi)^2\right], & \theta^- = \theta \\ +\infty, & \theta^- < \theta \\ -\infty, & \theta^- > \theta \end{cases} \tag{7}$$

*Proof.* See Appendix A. □

Recall that typically $\theta^- \neq \theta$ when $\mu \neq 0$. In this case, Eq. (6) shows that the CM/CT objective is independent of $\xi$, thus providing no signal of the data distribution and are therefore impossible to train correct consistency models. This directly refutes Argument (i). In contrast, when we set $\mu = 0$ to ensure $\theta^- = \theta$, Eq. (7) indicates that the gradient of the CM/CT objective, when scaled by $1/\Delta\sigma$, converges to the gradient of the mean squared error between $\theta$ and $\xi$. Optimizing this gradient consequently yields $\theta = \xi$, accurately learning the ground truth consistency function. This analysis is consistent with Argument (ii).

As illustrated in Fig. 2a, discarding EMA from the teacher network notably improves sample quality for CT across both LPIPS and squared $\ell_2$ metrics. The curves labeled "Improved" correspond to CT using the improved design outlined in Section 3.1. Setting $\mu(k) = 0$ for all training iteration $k$ effectively counters the sample quality degradation of LPIPS caused by the modifications in Section 3.1. Combining the strategies from Section 3.1 with a zero EMA for the teacher, we are able to match the sample quality of the original CT in Song et al. (2023) that necessitates LPIPS, by using simple squared $\ell_2$ metrics.

### 3.3 PSEUDO-HUBER METRIC FUNCTIONS

Using the methods from Sections 3.1 and 3.2, we are able to improve CT with squared $\ell_2$ metric, matching the original CT in Song et al. (2023) that utilizes LPIPS. Yet, as shown in Fig. 2a, LPIPS still maintains a significant advantage over traditional metric functions when the same improved techniques are in effect for all. To address this disparity, we adopt the Pseudo-Huber metric family (Charbonnier et al., 1997), defined as

$$d(\boldsymbol{x}, \boldsymbol{y}) = \sqrt{\|\boldsymbol{x} - \boldsymbol{y}\|_2^2 + c^2} - c, \tag{8}$$

where $c > 0$ is an adjustable constant. As depicted in Fig. 5a, Pseudo-Huber metrics smoothly bridge the $\ell_1$ and squared $\ell_2$ metrics, with $c$ determining the breadth of the parabolic section. In contrast to common metrics like $\ell_0$, $\ell_1$, and $\ell_\infty$, Pseudo-Huber metrics are continuously twice differentiable, and hence meet the theoretical requirement for CT outlined in Song et al. (2023).

Compared to the squared $\ell_2$ metric, the Pseudo-Huber metric is more robust to outliers as it imposes a smaller penalty for large errors than the squared $\ell_2$ metric does, yet behaves similarly for smaller errors. We posit that this added robustness can reduce variance during training. To validate this hypothesis, we examine the $\ell_2$ norms of parameter updates obtained from the Adam optimizer during the course of training for both squared $\ell_2$ and Pseudo-Huber metric functions, and summarize results in Fig. 5b. Our observations confirm that the Pseudo-Huber metric results in reduced variance relative to the squared $\ell_2$ metric, aligning with our hypothesis.

We evaluate the effectiveness of Pseudo-Huber metrics by training several consistency models with varying $c$ values on CIFAR-10 and comparing their sample quality with models trained using LPIPS and squared $\ell_2$ metrics. We incorporate improved techniques from Sections 3.1 and 3.2 for all metrics. Fig. 2 reveals that Pseudo-Huber metrics yield notably better sample quality than the squared $\ell_2$ metric. By increasing the overall size of $N(k)$—adjusting $s_0$ and $s_1$ from the standard values of 2 and 150 in Song et al. (2023) to our new values of 10 and 1280 (more in Section 3.4)—we for the first time surpass the performance of CT with LPIPS on equal footing using a traditional metric function that does not rely on learned feature representations. Furthermore, Fig. 2c indicates that $c = 0.03$ is optimal for CIFAR-10 images. We suggest that $c$ should scale linearly with $\|\boldsymbol{x} - \boldsymbol{y}\|_2$, and propose a heuristic of $c = 0.00054\sqrt{d}$ for images with $d$ dimensions. Empirically, we find this recommendation to work well on both CIFAR-10 and ImageNet $64 \times 64$ datasets.

### 3.4 IMPROVED CURRICULUM FOR TOTAL DISCRETIZATION STEPS

As mentioned in Section 3.2, CT's theoretical foundation holds asymptotically as $N \to \infty$. In practice, we have to select a finite $N$ for training consistency models, potentially introducing bias into the learning process. To understand the influence of $N$ on sample quality, we train a consistency model with improved techniques from Sections 3.1 to 3.3. Unlike Song et al. (2023), we use an exponentially increasing curriculum for the total discretization steps $N$, doubling $N$ after a set number of training iterations. Specifically, the curriculum is described by

$$N(k) = \min(s_0 2^{\lfloor \frac{k}{K'} \rfloor}, s_1) + 1, \quad K' = \left\lfloor \frac{K}{\log_2 \lfloor s_1/s_0 \rfloor + 1} \right\rfloor, \tag{9}$$

and its shape is labelled "Exp" in Fig. 3b. Here $s_0$ and $s_1$ control the minimum and maximum number of discretization steps, and $K$ is the total number of training iterations.

As revealed in Fig. 3a, the sample quality of consistency models improves predictably as $N$ increases. Importantly, FID scores relative to $N$ adhere to a precise power law until reaching saturation, after which further increases in $N$ yield diminishing benefits. As noted by Song et al. (2023), while larger $N$ can reduce bias in CT, they might increase variance. On the contrary, smaller $N$ reduces variance at the cost of higher bias. Based on Fig. 3a, we cap $N$ at 1281 in $N(k)$, which we empirically find to strike a good balance between bias and variance. In our experiments, we set $s_0$ and $s_1$ in discretization curriculums from their default values of 2 and 150 in Song et al. (2023) to 10 and 1280 respectively.

Aside from the exponential curriculum above, we also explore various shapes for $N(k)$ with the same $s_0 = 10$ and $s_1 = 1280$, including a constant function, the square root function from Song et al. (2023), a linear function, a square function, and a cosine function. The shapes of various curriculums are illustrated in Fig. 3b. As Fig. 3c demonstrates, the exponential curriculum yields the best sample quality for consistency models. Consequently, we adopt the exponential curriculum in Eq. (9) as our standard for setting $N(k)$ going forward.

### 3.5 IMPROVED NOISE SCHEDULES

Song et al. (2023) propose to sample a random $i$ from $\mathcal{U}[\![1, N-1]\!]$ and select $\sigma_i$ and $\sigma_{i+1}$ to compute the CT objective. Given that $\sigma_i = (\sigma_{\min}^{1/\rho} + \frac{i-1}{N-1}(\sigma_{\max}^{1/\rho} - \sigma_{\min}^{1/\rho}))^\rho$, this corresponds to sampling from the distribution $p(\log \sigma) = \sigma \frac{\sigma^{1/\rho - 1}}{\rho(\sigma_{\max}^{1/\rho} - \sigma_{\min}^{1/\rho})}$ as $N \to \infty$. As shown in Fig. 4a, this distribution exhibits a

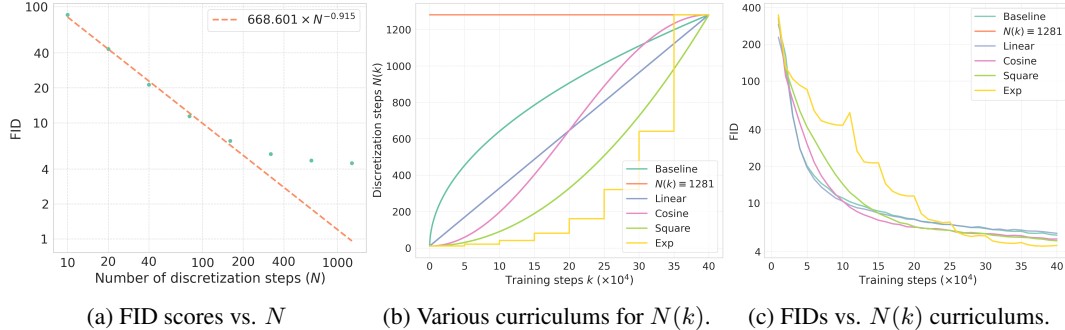

(a) FID scores vs. $N$  (b) Various curriculums for $N(k)$.  (c) FIDs vs. $N(k)$ curriculums.

Figure 3: (a) FID scores improve predictably as the number of discretization steps $N$ grows. (b) The shapes of various curriculums for total discretization steps $N(k)$. (c) The FID curves of various curriculums for discretization. All models are trained with improved techniques from Sections 3.1 to 3.3 with the only difference in discretization curriculums.

higher probability density for larger values of $\log \sigma$. This is at odds with the intuition that consistency losses at lower noise levels influence subsequent ones and cause error accumulation, so losses at lower noise levels should be given greater emphasis. Inspired by Karras et al. (2022), we address this by adopting a lognormal distribution to sample noise levels, setting a mean of -1.1 and a standard deviation of 2.0. As illustrated in Fig. 4a, this lognormal distribution assigns significantly less weight to high noise levels. Moreover, it also moderates the emphasis on smaller noise levels. This is helpful because learning is easier at smaller noise levels due to the inductive bias in our parameterization of the consistency model to meet the boundary condition.

For practical implementation, we sample noise levels in the set $\{\sigma_1, \sigma_2, \cdots, \sigma_N\}$ according to a discretized lognormal distribution defined as

$$p(\sigma_i) \propto \operatorname{erf}\left(\frac{\log(\sigma_{i+1}) - P_{\text{mean}}}{\sqrt{2}P_{\text{std}}}\right) - \operatorname{erf}\left(\frac{\log(\sigma_i) - P_{\text{mean}}}{\sqrt{2}P_{\text{std}}}\right), \tag{10}$$

where $P_{\text{mean}} = -1.1$ and $P_{\text{std}} = 2.0$. As depicted in Fig. 4b, this lognormal noise schedule significantly improves the sample quality of consistency models.

## 4 PUTTING IT TOGETHER

Combining all the improved techniques from Sections 3.1 to 3.5, we employ CT to train several consistency models on CIFAR-10 and ImageNet $64 \times 64$ and benchmark their performance with competing methods in the literature. We evaluate sample quality using FID (Heusel et al., 2017), Inception score (Salimans et al., 2016), and Precision/Recall (Kynkäänniemi et al., 2019). For best performance, we use a larger batch size and an increased EMA decay rate for the student network in CT across all models. The model architectures are based on Score SDE (Song et al., 2021) for CIFAR-10 and ADM (Dhariwal & Nichol, 2021) for ImageNet $64 \times 64$. We also explore deeper variants of these architectures by doubling the model depth. We call our method **iCT** which stands for "improved consistency training", and the deeper variants **iCT-deep**. We summarize our results in Tables 2 and 3 and provide uncurated samples from both iCT and iCT-deep in Figs. 6 to 9. More experimental details and results are provided in Appendix B.

It is important to note that we exclude methods based on FastGAN (Liu et al., 2020; Sauer et al., 2021) or StyleGAN-XL (Sauer et al., 2022) from our comparison, because both utilize ImageNet pre-trained feature extractors in their discriminators. As noted by Kynkäänniemi et al. (2023), this can skew FIDs and lead to inflated sample quality. Methods based on LPIPS suffer from similar issues, as LPIPS is also pre-trained on ImageNet. We include these methods in Tables 2 and 3 for completeness, but we do not consider them as direct competitors to iCT or iCT-deep methods.

Several key observations emerge from Tables 2 and 3. First, iCT methods *surpass previous diffusion distillation approaches in both one-step and two-step generation* on CIFAR-10 and ImageNet $64 \times 64$, all while circumventing the need for training diffusion models. Secondly, iCT models demonstrate

Table 2: Comparing the quality of unconditional samples on CIFAR-10.

| METHOD | NFE (↓) | FID (↓) | IS (↑) |
|---|---|---|---|
| **Fast samplers & distillation for diffusion models** | | | |
| DDIM (Song et al., 2020) | 10 | 13.36 | |
| DPM-solver-fast (Lu et al., 2022) | 10 | 4.70 | |
| 3-DEIS (Zhang & Chen, 2022) | 10 | 4.17 | |
| UniPC (Zhao et al., 2023) | 10 | 3.87 | |
| Knowledge Distillation (Luhman & Luhman, 2021) | 1 | 9.36 | |
| DFNO (LPIPS) (Zheng et al., 2022) | 1 | 3.78 | |
| 2-Rectified Flow (+distill) (Liu et al., 2022) | 1 | 4.85 | 9.01 |
| TRACT (Berthelot et al., 2023) | 1 | 3.78 | |
| | 2 | 3.32 | |
| Diff-Instruct (Luo et al., 2023) | 1 | 4.53 | 9.89 |
| PD* (Salimans & Ho, 2022) | 1 | 8.34 | 8.69 |
| | 2 | 5.58 | 9.05 |
| CD (LPIPS) (Song et al., 2023) | 1 | 3.55 | 9.48 |
| | 2 | 2.93 | 9.75 |
| **Direct Generation** | | | |
| Score SDE (Song et al., 2021) | 2000 | 2.38 | 9.83 |
| Score SDE (deep) (Song et al., 2021) | 2000 | 2.20 | 9.89 |
| DDPM (Ho et al., 2020) | 1000 | 3.17 | 9.46 |
| LSGM (Vahdat et al., 2021) | 147 | 2.10 | |
| PFGM (Xu et al., 2022) | 110 | 2.35 | 9.68 |
| EDM* (Karras et al., 2022) | 35 | 2.04 | 9.84 |
| EDM-G++ (Kim et al., 2023) | 35 | 1.77 | |
| IGEBM (Du & Mordatch, 2019) | 60 | 40.6 | 6.02 |
| NVAE (Vahdat & Kautz, 2020) | 1 | 23.5 | 7.18 |
| Glow (Kingma & Dhariwal, 2018) | 1 | 48.9 | 3.92 |
| Residual Flow (Chen et al., 2019) | 1 | 46.4 | |
| BigGAN (Brock et al., 2019) | 1 | 14.7 | 9.22 |
| StyleGAN2 (Karras et al., 2020b) | 1 | 8.32 | 9.21 |
| StyleGAN2-ADA (Karras et al., 2020a) | 1 | 2.92 | 9.83 |
| CT (LPIPS) (Song et al., 2023) | 1 | 8.70 | 8.49 |
| | 2 | 5.83 | 8.85 |
| **iCT (ours)** | 1 | 2.83 | 9.54 |
| | 2 | 2.46 | 9.80 |
| **iCT-deep (ours)** | 1 | 2.51 | 9.76 |
| | 2 | 2.24 | 9.89 |

Table 3: Comparing the quality of class-conditional samples on ImageNet $64 \times 64$.

| METHOD | NFE (↓) | FID (↓) | Prec. (↑) | Rec. (↑) |
|---|---|---|---|---|
| **Fast samplers & distillation for diffusion models** | | | | |
| DDIM (Song et al., 2020) | 50 | 13.7 | 0.65 | 0.56 |
| | 10 | 18.3 | 0.60 | 0.49 |
| DPM solver (Lu et al., 2022) | 10 | 7.93 | | |
| | 20 | 3.42 | | |
| DEIS (Zhang & Chen, 2022) | 10 | 6.65 | | |
| | 20 | 3.10 | | |
| DFNO (LPIPS) (Zheng et al., 2022) | 1 | 7.83 | | 0.61 |
| TRACT (Berthelot et al., 2023) | 1 | 7.43 | | |
| | 2 | 4.97 | | |
| BOOT (Gu et al., 2023) | 1 | 16.3 | 0.68 | 0.36 |
| Diff-Instruct (Luo et al., 2023) | 1 | 5.57 | | |
| PD* (Salimans & Ho, 2022) | 1 | 15.39 | 0.59 | 0.62 |
| | 2 | 8.95 | 0.63 | 0.65 |
| | 4 | 6.77 | 0.66 | 0.65 |
| PD (LPIPS) (Song et al., 2023) | 1 | 7.88 | 0.66 | 0.63 |
| | 2 | 5.74 | 0.67 | 0.65 |
| | 4 | 4.92 | 0.68 | 0.65 |
| CD (LPIPS) (Song et al., 2023) | 1 | 6.20 | 0.68 | 0.63 |
| | 2 | 4.70 | 0.69 | 0.64 |
| | 3 | 4.32 | 0.70 | 0.64 |
| **Direct Generation** | | | | |
| RIN (Jabri et al., 2023) | 1000 | 1.23 | | |
| DDPM (Ho et al., 2020) | 250 | 11.0 | 0.67 | 0.58 |
| iDDPM (Nichol & Dhariwal, 2021) | 250 | 2.92 | 0.74 | 0.62 |
| ADM (Dhariwal & Nichol, 2021) | 250 | 2.07 | 0.74 | 0.63 |
| EDM (Karras et al., 2022) | 511 | 1.36 | | |
| EDM* (Heun) (Karras et al., 2022) | 79 | 2.44 | 0.71 | 0.67 |
| BigGAN-deep (Brock et al., 2019) | 1 | 4.06 | 0.79 | 0.48 |
| CT (LPIPS) (Song et al., 2023) | 1 | 13.0 | 0.71 | 0.47 |
| | 2 | 11.1 | 0.69 | 0.56 |
| **iCT (ours)** | 1 | 4.02 | 0.70 | 0.63 |
| | 2 | 3.20 | 0.73 | 0.63 |
| **iCT-deep (ours)** | 1 | 3.25 | 0.72 | 0.63 |
| | 2 | 2.77 | 0.74 | 0.62 |

Most results for existing methods are taken from a previous paper, except for those marked with *, which are from our own re-implementation.

sample quality comparable to many leading generative models, including diffusion models and GANs. For instance, with one-step generation, iCT-deep obtains FIDs of 2.51 and 3.25 for CIFAR-10 and ImageNet respectively, whereas DDPMs (Ho et al., 2020) necessitate thousands of sampling steps to reach FIDs of 3.17 and 11.0 (result taken from Gu et al. (2023)) on both datasets. The one-step FID for iCT already exceeds that of StyleGAN-ADA (Karras et al., 2020b) on CIFAR-10, and that of BigGAN-deep (Brock et al., 2019) on ImageNet $64 \times 64$, let alone iCT-deep models. For two-step generation, iCT-deep records an FID of 2.24, matching Score SDE in Song et al. (2021), a diffusion model with an identical architecture but demands 2000 sampling steps for an FID of 2.20. Lastly, iCT methods show improved recall than CT (LPIPS) in Song et al. (2023) and BigGAN-deep, indicating better diversity and superior mode coverage.

## 5 CONCLUSION

Our improved techniques for CT have successfully addressed its previous limitations, surpassing the performance of CD in generating high-quality samples without relying on LPIPS. We examined the impact of weighting functions, noise embeddings, and dropout. By removing EMA for teacher networks, adopting Pseudo-Huber losses in lieu of LPIPS, combined with a new curriculum for discretization and noise sampling schedule, we have achieved unprecedented FID scores for consistency models on both CIFAR-10 and ImageNet $64 \times 64$ datasets. Remarkably, these results outpace previous CT methods by a considerable margin, surpass previous few-step diffusion distillation techniques, and challenge the sample quality of leading diffusion models and GANs.

ACKNOWLEDGEMENTS

We would like to thank Alex Nichol, Allan Jabri, Ishaan Gulrajani, Jakub Pachocki, Mark Chen and Ilya Sutskever for discussions and support throughout this project.

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

## A PROOFS

**Proposition 1.** *Given the notations introduced in Section 3.2, and using the uniform weighting function $\lambda(\sigma) = 1$ along with the squared $\ell_2$ metric, we have*

$$\lim_{N \to \infty} \mathcal{L}^N(\theta, \theta^-) = \lim_{N \to \infty} \mathcal{L}_{CT}^N(\theta, \theta^-) = \mathbb{E}\left[\left(1 - \frac{\sigma_{min}}{\sigma_i}\right)^2 (\theta - \theta^-)^2\right] \quad \text{if } \theta^- \neq \theta \tag{11}$$

$$\lim_{N \to \infty} \frac{1}{\Delta\sigma} \frac{\mathrm{d}\mathcal{L}^N(\theta, \theta^-)}{\mathrm{d}\theta} = \begin{cases} \frac{\mathrm{d}}{\mathrm{d}\theta} \mathbb{E}\left[\frac{\sigma_{min}}{\sigma_i^2}\left(1 - \frac{\sigma_{min}}{\sigma_i}\right)(\theta - \xi)^2\right], & \theta^- = \theta \\ +\infty, & \theta^- < \theta \\ -\infty, & \theta^- > \theta \end{cases} \tag{12}$$

*Proof.* Since $\lambda(\sigma) \equiv 1$ and $d(x, y) = (x - y)^2$, we can write down the CM and CT objectives as $\mathcal{L}^N(\theta, \theta^-) = \mathbb{E}[(f_\theta(x_{\sigma_{i+1}}, \sigma_{i+1}) - f_{\theta^-}(\breve{x}_{\sigma_i}, \sigma_i))^2]$ and $\mathcal{L}_{CT}^N(\theta, \theta^-) = \mathbb{E}[(f_\theta(x_{\sigma_{i+1}}, \sigma_{i+1}) - f_{\theta^-}(\breve{x}_{\sigma_i}, \sigma_i))^2]$ respectively. Since $p_{\text{data}}(x) = \delta(x - \xi)$, we have $p_\sigma(x) = \mathcal{N}(x \mid \xi, \sigma^2)$, and therefore $\nabla \log p_\sigma(x) = -\frac{x - \xi}{\sigma^2}$. According to the definition of $\breve{x}_{\sigma_i}$ and $x_{\sigma_{i+1}} = \xi + \sigma_{i+1}z$, we have

$$\begin{aligned} \breve{x}_{\sigma_i} &= x_{\sigma_{i+1}} - (\sigma_i - \sigma_{i+1})\sigma_{i+1}\nabla \log p(x_{\sigma_{i+1}}, \sigma_{i+1}) \\ &= x_{\sigma_{i+1}} + (\sigma_i - \sigma_{i+1})\sigma_{i+1}\frac{x_{\sigma_{i+1}} - \xi}{\sigma_{i+1}^2} \\ &= x_{\sigma_{i+1}} + (\sigma_i - \sigma_{i+1})z \\ &= \xi + \sigma_{i+1}z + (\sigma_i - \sigma_{i+1})z \\ &= \xi + \sigma_i z \\ &= \breve{x}_{\sigma_i}. \end{aligned}$$

As a result, the CM and CT objectives are exactly the same, that is, $\mathcal{L}^N(\theta, \theta^-) = \mathcal{L}_{CT}^N(\theta, \theta^-)$. Recall that the consistency model $f_\theta(x, \sigma)$ is defined as $f_\theta(x, \sigma) = \frac{\sigma_{min}}{\sigma}x + \left(1 - \frac{\sigma_{min}}{\sigma}\right)\theta$, so we have $f_\theta(x_\sigma, \sigma) = \sigma_{min}z + \frac{\sigma_{min}}{\sigma}\xi + \left(1 - \frac{\sigma_{min}}{\sigma}\right)\theta$. Now, let us focus on the CM objective

$$\begin{aligned} \mathcal{L}^N(\theta, \theta^-) &= \mathbb{E}[(f_\theta(x_{\sigma_{i+1}}, \sigma_{i+1}) - f_{\theta^-}(\breve{x}_{\sigma_i}, \sigma_i))^2] \\ &= \mathbb{E}[(f_\theta(x_{\sigma_{i+1}}, \sigma_{i+1}) - f_{\theta^-}(\breve{x}_{\sigma_i}, \sigma_i))^2] \\ &= \mathbb{E}\left[\left(\frac{\sigma_{\min}}{\sigma_{i+1}}\xi + \left(1 - \frac{\sigma_{\min}}{\sigma_{i+1}}\right)\theta - \frac{\sigma_{\min}}{\sigma_i}\xi - \left(1 - \frac{\sigma_{\min}}{\sigma_i}\right)\theta^-\right)^2\right] \\ &= \mathbb{E}\left[\left(\frac{\sigma_{\min}}{\sigma_i + \Delta\sigma}\xi + \left(1 - \frac{\sigma_{\min}}{\sigma_i + \Delta\sigma}\right)\theta - \frac{\sigma_{\min}}{\sigma_i}\xi - \left(1 - \frac{\sigma_{\min}}{\sigma_i}\right)\theta^-\right)^2\right], \end{aligned}$$

where $\Delta\sigma = \frac{\sigma_{\max} - \sigma_{\min}}{N - 1}$, because $\sigma_i = \sigma_{\min} + \frac{i-1}{N-1}(\sigma_{\max} - \sigma_{\min})$. By taking the limit $N \to \infty$, we have $\Delta\sigma \to 0$, and therefore

$$\lim_{N \to \infty} \mathcal{L}^N(\theta, \theta^-)$$

$$= \lim_{\Delta\sigma \to 0} \mathbb{E}\left[\left(\frac{\sigma_{\min}}{\sigma_i + \Delta\sigma}\xi + \left(1 - \frac{\sigma_{\min}}{\sigma_i + \Delta\sigma}\right)\theta - \frac{\sigma_{\min}}{\sigma_i}\xi - \left(1 - \frac{\sigma_{\min}}{\sigma_i}\right)\theta^-\right)^2\right]$$

$$= \lim_{\Delta\sigma \to 0} \mathbb{E}\left[\left(\frac{\sigma_{\min}}{\sigma_i}\left(1 - \frac{\Delta\sigma}{\sigma_i}\right)\xi + \left(1 - \frac{\sigma_{\min}}{\sigma_i + \Delta\sigma}\right)\theta - \frac{\sigma_{\min}}{\sigma_i}\xi - \left(1 - \frac{\sigma_{\min}}{\sigma_i}\right)\theta^-\right)^2\right] + o(\Delta\sigma)$$

$$= \lim_{\Delta\sigma \to 0} \mathbb{E}\left[\left(-\frac{\sigma_{\min}\Delta\sigma}{\sigma_i^2}\xi + \left(1 - \frac{\sigma_{\min}}{\sigma_i + \Delta\sigma}\right)\theta - \left(1 - \frac{\sigma_{\min}}{\sigma_i}\right)\theta^-\right)^2\right] + o(\Delta\sigma)$$

$$= \lim_{\Delta\sigma \to 0} \mathbb{E}\left[\left(-\frac{\sigma_{\min}\Delta\sigma}{\sigma_i^2}\xi + \left(1 - \frac{\sigma_{\min}}{\sigma_i}\left(1 - \frac{\Delta\sigma}{\sigma_i}\right)\right)\theta - \left(1 - \frac{\sigma_{\min}}{\sigma_i}\right)\theta^-\right)^2\right] + o(\Delta\sigma).$$

Suppose $\theta^- \neq \theta$, we have

$$
\lim_{N \to \infty} \mathcal{L}^N(\theta, \theta^-)
$$

$$
= \lim_{\Delta\sigma \to 0} \mathbb{E}\left[\left(-\frac{\sigma_{\min}\Delta\sigma}{\sigma_i^2}\xi + \left(1 - \frac{\sigma_{\min}}{\sigma_i}\left(1 - \frac{\Delta\sigma}{\sigma_i}\right)\right)\theta - \left(1 - \frac{\sigma_{\min}}{\sigma_i}\right)\theta^-\right)^2\right] + o(\Delta\sigma)
$$

$$
= \lim_{\Delta\sigma \to 0} \mathbb{E}\left[\left(1 - \frac{\sigma_{\min}}{\sigma_i}\right)^2(\theta - \theta^-)^2\right] + o(\Delta\sigma)
$$

$$
= \mathbb{E}\left[\left(1 - \frac{\sigma_{\min}}{\sigma_i}\right)^2(\theta - \theta^-)^2\right],
$$

which proves our first statement in the proposition.

Now, let's consider $\nabla_\theta \mathcal{L}^N(\theta, \theta^-)$. It has the following form

$$
\nabla_\theta \mathcal{L}^N(\theta, \theta^-) = 2\mathbb{E}\left[\left(\frac{\sigma_{\min}}{\sigma_{i+1}}\xi + \left(1 - \frac{\sigma_{\min}}{\sigma_{i+1}}\right)\theta - \frac{\sigma_{\min}}{\sigma_i}\xi - \left(1 - \frac{\sigma_{\min}}{\sigma_i}\right)\theta^-\right)\left(1 - \frac{\sigma_{\min}}{\sigma_{i+1}}\right)\right].
$$

As $N \to \infty$ and $\Delta\sigma \to 0$, we have

$$
\lim_{N \to \infty} \nabla_\theta \mathcal{L}^N(\theta, \theta^-)
$$

$$
= \lim_{\Delta\sigma \to 0} 2\mathbb{E}\left[-\frac{\sigma_{\min}\Delta\sigma}{\sigma_i^2}\xi + \left(1 - \frac{\sigma_{\min}}{\sigma_i}\left(1 - \frac{\Delta\sigma}{\sigma_i}\right)\right)\theta - \left(1 - \frac{\sigma_{\min}}{\sigma_i}\right)\theta^-\right]\left(1 - \frac{\sigma_{\min}}{\sigma_i}\right)
$$

$$
= \begin{cases} \lim_{\Delta\sigma \to 0} 2\mathbb{E}\left[-\frac{\sigma_{\min}\Delta\sigma}{\sigma_i^2}\xi + \frac{\sigma_{\min}\Delta\sigma}{\sigma_i^2}\theta\right]\left(1 - \frac{\sigma_{\min}}{\sigma_i}\right), & \theta^- = \theta \\ 2\mathbb{E}\left[\left(1 - \frac{\sigma_{\min}}{\sigma}\right)^2(\theta - \theta^-)\right], & \theta^- \neq \theta \end{cases}
$$

$$
= \begin{cases} \lim_{\Delta\sigma \to 0} 2\mathbb{E}\left[\frac{\sigma_{\min}\Delta\sigma}{\sigma_i^2}(\theta - \xi)\right]\left(1 - \frac{\sigma_{\min}}{\sigma_i}\right), & \theta^- = \theta \\ 2\mathbb{E}\left[\left(1 - \frac{\sigma_{\min}}{\sigma_i}\right)^2(\theta - \theta^-)\right], & \theta^- \neq \theta. \end{cases} \tag{13}
$$

Now it becomes obvious from Eq. (13) that when $\theta^- = \theta$, we have

$$
\lim_{N \to \infty} \frac{1}{\Delta\sigma} \nabla_\theta \mathcal{L}^N(\theta, \theta^-) = \lim_{\Delta\sigma \to 0} 2\mathbb{E}\left[\frac{\sigma_{\min}}{\sigma_i^2}(\theta - \xi)\right]\left(1 - \frac{\sigma_{\min}}{\sigma_i}\right)
$$

$$
= 2\mathbb{E}\left[\frac{\sigma_{\min}}{\sigma_i^2}(\theta - \xi)\right]\left(1 - \frac{\sigma_{\min}}{\sigma_i}\right)
$$

$$
= \frac{\mathrm{d}}{\mathrm{d}\theta}\mathbb{E}\left[\frac{\sigma_{\min}}{\sigma_i^2}\left(1 - \frac{\sigma_{\min}}{\sigma_i}\right)(\theta - \xi)^2\right].
$$

Moreover, we can deduce from Eq. (13) that

$$
\lim_{N \to \infty} \frac{1}{\Delta\sigma} \nabla_\theta \mathcal{L}^N(\theta, \theta^-) = \begin{cases} +\infty, & \theta > \theta^- \\ -\infty, & \theta < \theta^- \end{cases},
$$

which concludes the proof. $\qquad\square$

## B  ADDITIONAL EXPERIMENTAL DETAILS AND RESULTS

**Model architecture**  Unless otherwise noted, we use the NCSN++ architecture (Song et al., 2021) on CIFAR-10, and the ADM architecture (Dhariwal & Nichol, 2021) on ImageNet $64 \times 64$. For iCT-deep models in Tables 2 and 3, we double the depth of base architectures by increasing the number of residual blocks per resolution from 4 and 3 to 8 and 6 for CIFAR-10 and ImageNet $64 \times 64$ respectively. We use a dropout rate of 0.3 for all consistency models on CIFAR-10. For ImageNet $64 \times 64$, we use a dropout rate of 0.2, but only apply them to convolutional layers whose the feature map resolution is smaller or equal to $16 \times 16$, following the configuration in Hoogeboom et al. (2023). We also found that AdaGN introduced in Dhariwal & Nichol (2021) hurts consistency training and opt to remove it for our ImageNet $64 \times 64$ experiments. All models on CIFAR-10 are unconditional, and all models on ImageNet $64 \times 64$ are conditioned on class labels.

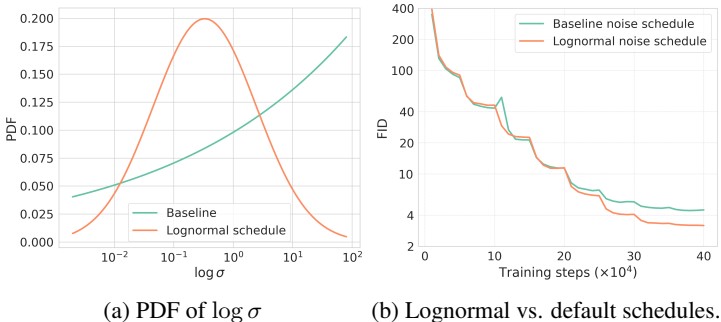

(a) PDF of $\log \sigma$  (b) Lognormal vs. default schedules.

Figure 4: The PDF of $\log \sigma$ indicates that the default noise schedule in Song et al. (2023) assigns more weight to larger values of $\log \sigma$, corrected by our lognormal schedule. We compare the FID scores of CT using both the lognormal noise schedule and the original one, where both models incorporate the improved techniques in Sections 3.1 to 3.4.

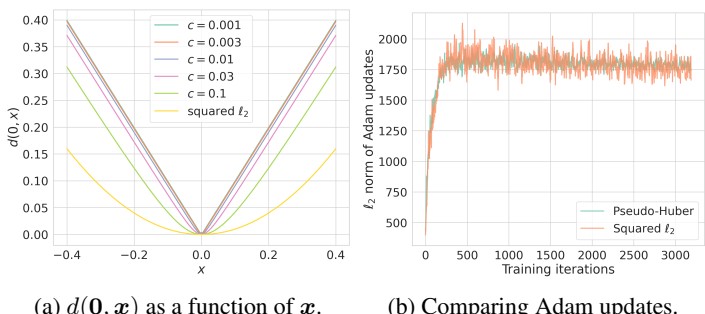

(a) $d(\mathbf{0}, \boldsymbol{x})$ as a function of $\boldsymbol{x}$.  (b) Comparing Adam updates.

Figure 5: (a) The shapes of various metric functions. (b) The $\ell_2$ norms of parameter updates in Adam optimizer. Curves are rescaled to have the same mean. The Pseudo-Huber metric has lower variance compared to the squared $\ell_2$ metric.

**Training**  We train all models with the RAdam optimizer (Liu et al., 2019) using learning rate 0.0001. All CIFAR-10 models are trained for 400,000 iterations, whereas ImageNet $64 \times 64$ models are trained for 800,000 iterations. For CIFAR-10 models in Section 3, we use batch size 512 and EMA decay rate 0.9999 for the student network. For iCT and iCT-deep models in Table 2, we use batch size 1024 and EMA decay rate of 0.99993 for CIFAR-10 models, and batch size 4096 and EMA decay rate 0.99997 for ImageNet $64 \times 64$ models. All models are trained on a cluster of Nvidia A100 GPUs.

**Pseudo-Huber losses and variance reduction**  In Fig. 5, we provide additional analysis for the Pseudo-Huber metric proposed in Section 3.3. We show the shapes of squared $\ell_2$ metric, as well as Pseudo-Huber losses with various values of $c$ in Fig. 5a, illustrating that Pseudo-Huber losses smoothly interpolates between the $\ell_1$ and squared $\ell_2$ metrics. In Fig. 5b, we plot the $\ell_2$ norms of parameter updates retrieved from the Adam optimizer for models trained with squared $\ell_2$ and Pseudo-Huber metrics. We observe that the Pseudo-Huber metric has lower variance compared to the squared $\ell_2$ metric, which is consistent with our hypothesis in Section 3.3.

**Samples**  We provide additional uncurated samples from iCT and iCT-deep models on both CIFAR-10 and ImageNet $64 \times 64$. See Figs. 6 to 9. For two-step sampling, the intermediate noise level $\sigma_{i_2}$ is 0.821 for CIFAR-10 and 1.526 for ImageNet $64 \times 64$ when using iCT. When employing iCT-deep, $\sigma_{i_2}$ is 0.661 for CIFAR-10 and 0.973 for ImageNet $64 \times 64$.

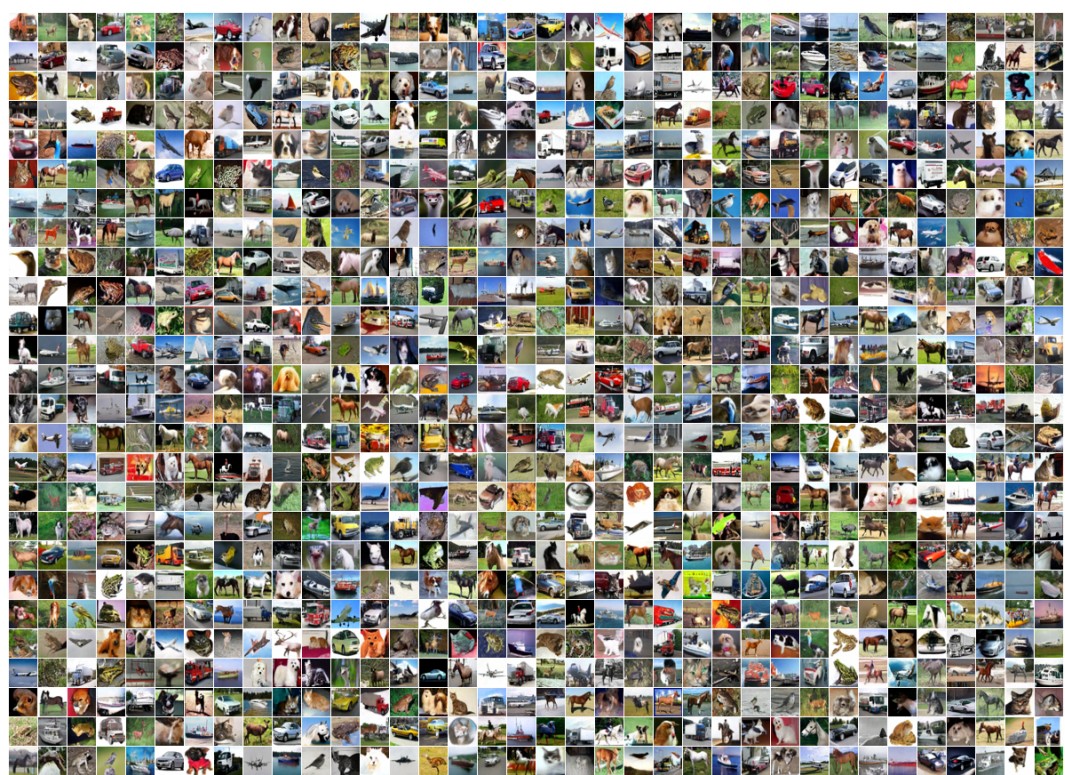

(a) One-step samples from the iCT model on CIFAR-10 (FID = 2.83).

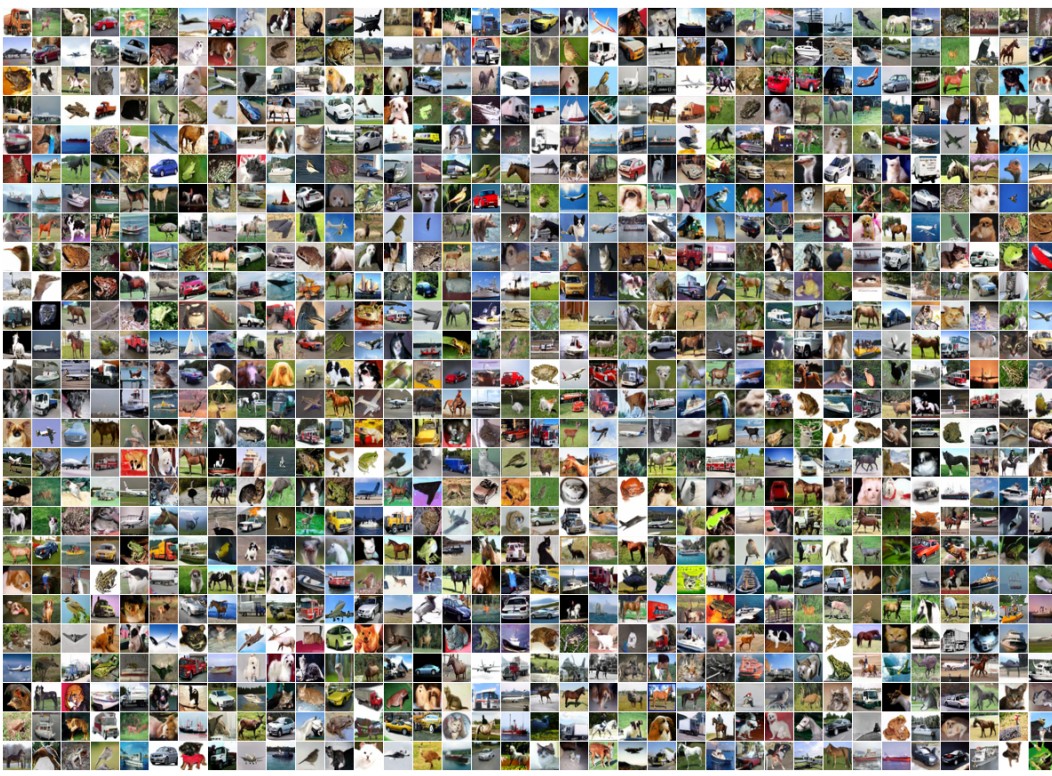

(b) Two-step samples from the iCT model on CIFAR-10 (FID = 2.46).

Figure 6: Uncurated samples from iCT models on CIFAR-10. All corresponding samples use the same initial noise.

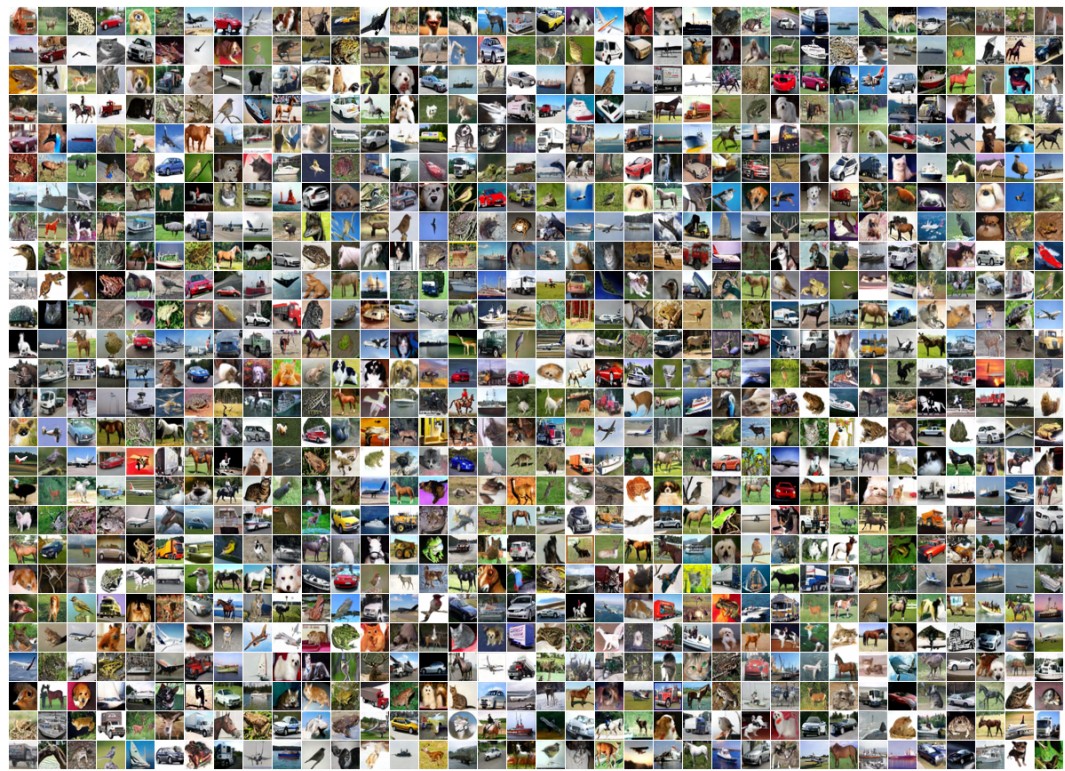

(a) One-step samples from the iCT-deep model on CIFAR-10 (FID = 2.51).

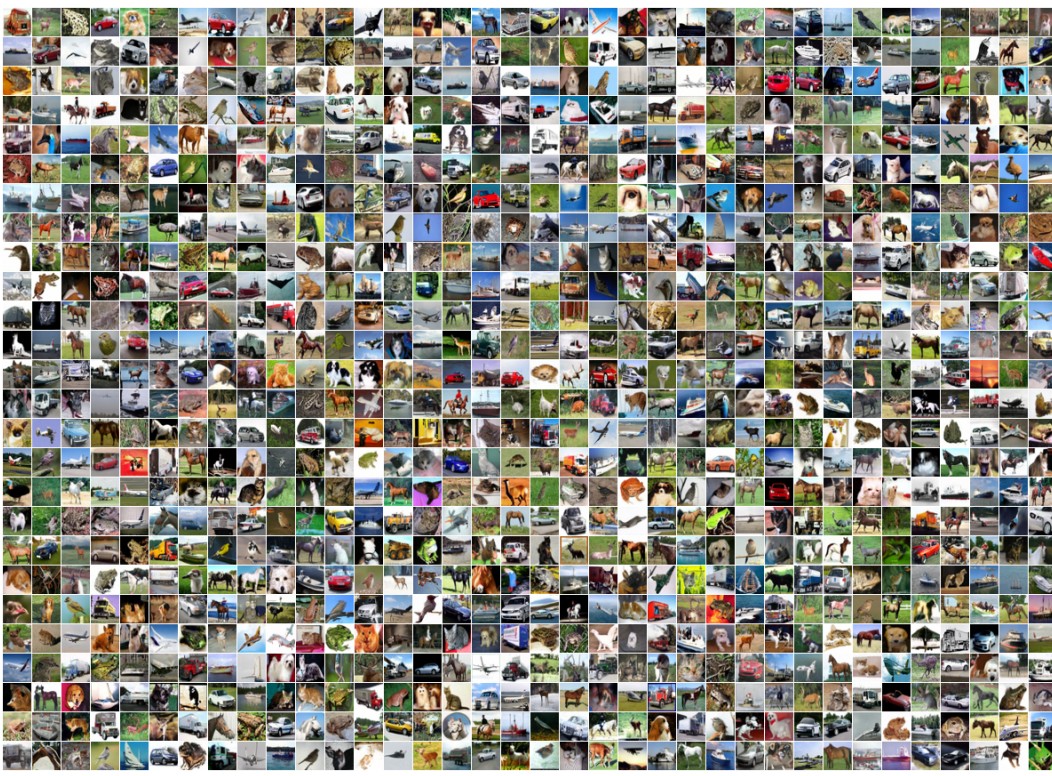

(b) Two-step samples from the iCT-deep model on CIFAR-10 (FID = 2.24).

Figure 7: Uncurated samples from iCT-deep models on CIFAR-10. All corresponding samples use the same initial noise.

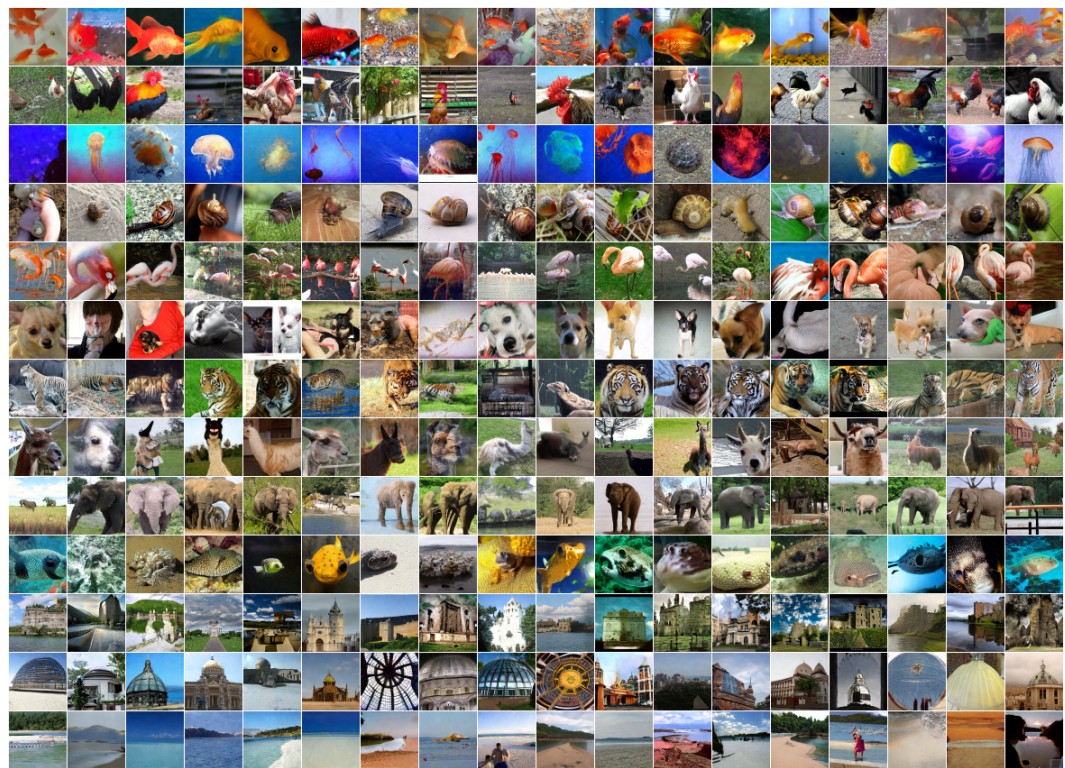

(a) One-step samples from the iCT model on ImageNet $64 \times 64$ (FID = 4.02).

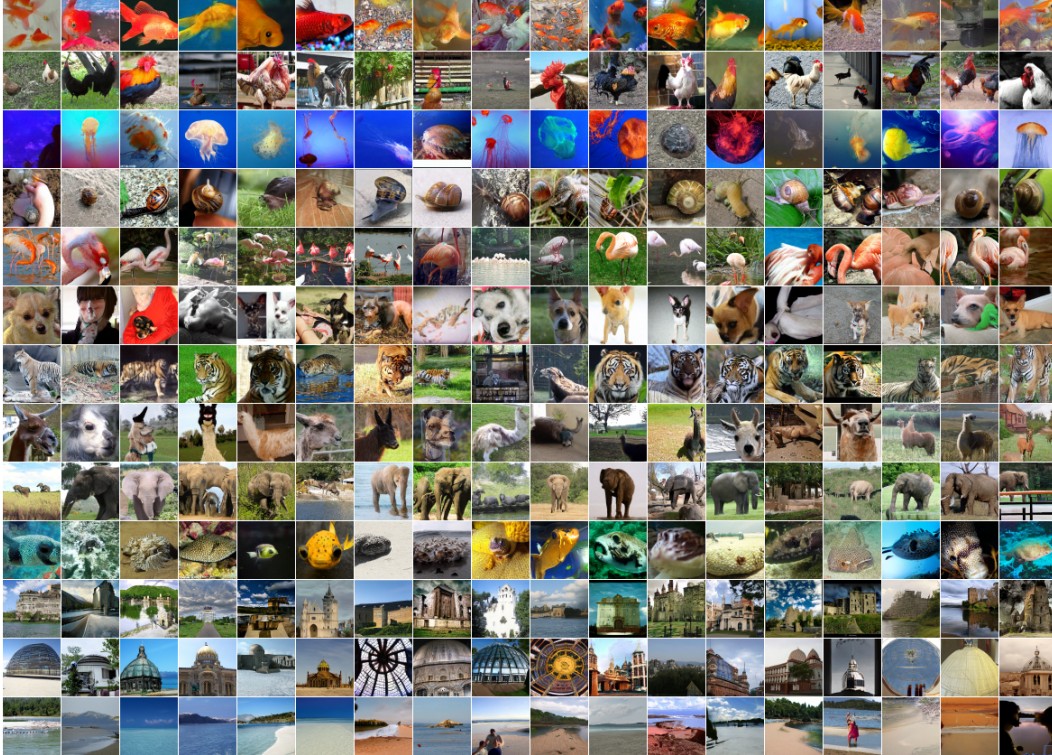

(b) Two-step samples from the iCT model on ImageNet $64 \times 64$ (FID = 3.20).

Figure 8: Uncurated samples from iCT models on ImageNet $64 \times 64$. All corresponding samples use the same initial noise.

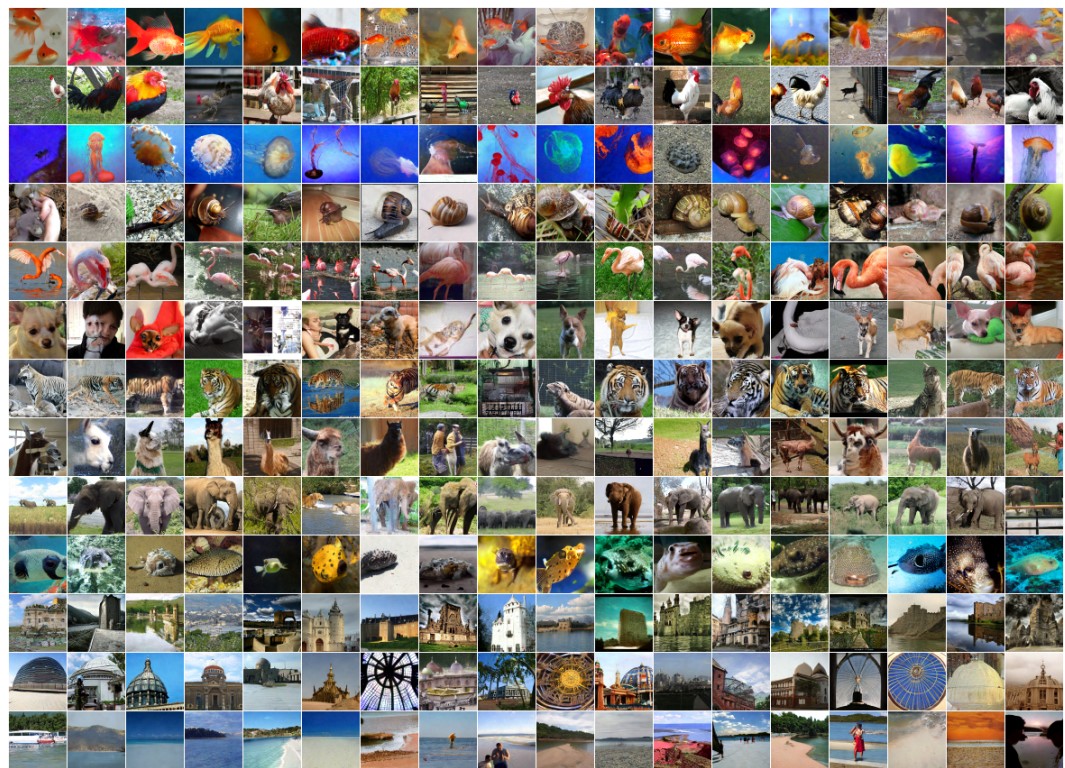

(a) One-step samples from the iCT-deep model on ImageNet $64 \times 64$ (FID = 3.25).

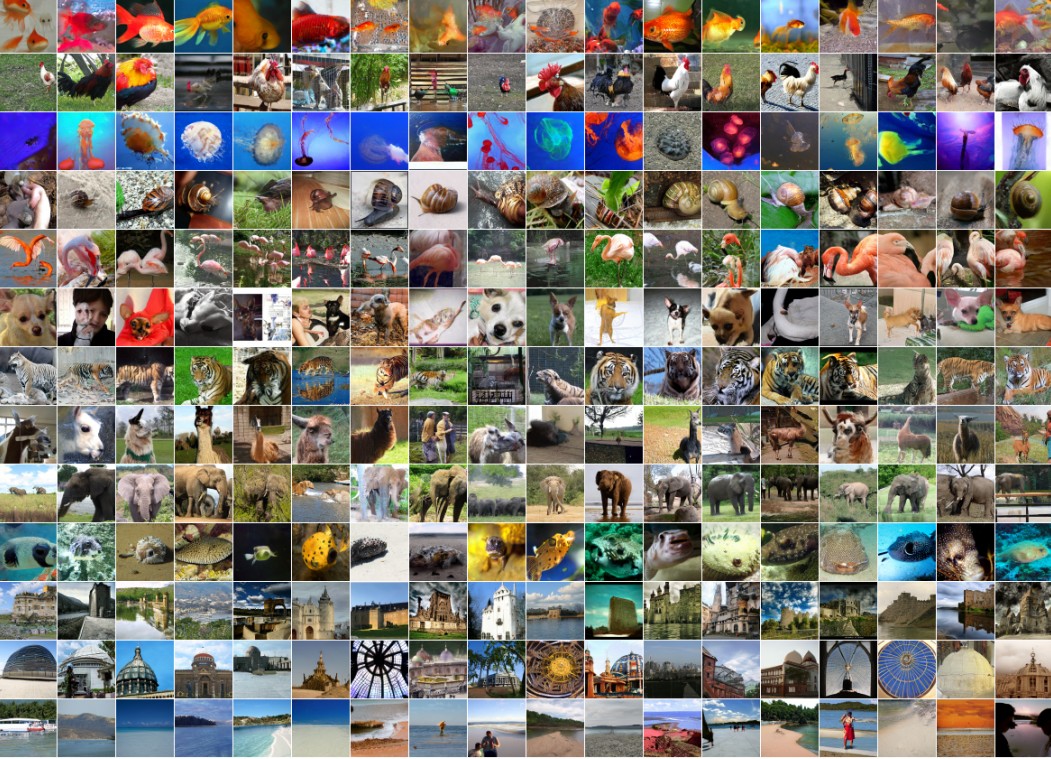

(b) Two-step samples from the iCT-deep model on ImageNet $64 \times 64$ (FID = 2.77).

Figure 9: Uncurated samples from iCT-deep models on ImageNet $64 \times 64$. All corresponding samples use the same initial noise.

