# OpenReview forum: "Improved Techniques for Training Consistency Models"
_ICLR.cc/2024/Conference — ICLR 2024 oral_

### Official Review · Reviewer_m7Dp · 2023-10-26

**Soundness:** 3 good
**Presentation:** 3 good
**Contribution:** 3 good
**Rating:** 8
**Confidence:** 4

**Summary:**

Recently, consistency models were proposed. They can be used to either distill diffusion models into single- or few-step samplers (consistency distillation), or they also enable direct training of single- or few-step consistency models (consistency training). Previous work showed strong results primarily for the distillation setting. In this work the authors focus on the consistency training setting and propose multiple techniques to improve the performance of consistency training. For instance, the paper avoids the questionable use of LPIPS-based consistency matching, and employs a more general Huber loss. The authors also remove the exponential moving average and point out errors in previous theoretical analyses. Moreover, a new loss weighting function is proposed, the noise level embeddings are carefully analyzed and improved, the use of dropout is studied, and the discretization step curriculum is improved. The paper thoroughly ablates and analyzes all improvements and then evaluates the model on popular benchmarks, CIFAR10 and ImageNet64. It achieves very strong performance, significantly outperforming both previous consistency models and diffusion model distillation methods. The improved consistency models perform almost as good as state-of-the-art diffusion models and GANs, while being trained directly and only requiring a single or two synthesis steps.

**Strengths:**

The paper has several strengths:
- The paper is well-written and easy to follow (good **clarity**).
- The paper does not present a fundamentally new method (moderate **novelty**). However, it does significantly improve consistency models, a very novel and very promising class of generative models that enables single- or few-step sampling, in contrast to, for instance, diffusion models. I expect the insights provided by the paper to be used in follow-up work and I think that consistency models will find wide usage, too. This makes this work very **significant**
- The paper presents thorough ablations and analyses of the proposed tricks and innovations. It is generally of high **quality**.
- The final experimental results obtained when combining all modifications are very strong, and thoroughly compared to many and appropriate baselines.

**Weaknesses:**

I think the paper has no major weaknesses. However, there would be opportunities to further improve the work:
- I am wondering how scalable consistency models are with the proposed modifications. Can one train, for instance, text-to-image consistency models? Or how about training on higher-resolution images?
- The new consistency models do not require LPIPS losses anymore and are thereby more general in that they are not limited to image synthesis anymore. It would be interesting to validate that consistency models can also be successfully trained on non-image data (e.g. audio, graphs, 3D, video, etc).

Conclusion: Overall, I think this is a good and solid paper. It significantly improves consistency models, a very promising new class of generative models, and the paper is overall well-written and of high quality. Hence, I recommend acceptance.

**Questions:**

I only have some minor questions:
- When the paper defines the ground truth consistency function in Section 3.2, is there a typo in the ground truth consistency function? Should it be $\xi$ instead of $\mu$ in the ground truth consistency function $f^{*}$ for the single data point data distribution?
- What are $s_0$ and $s_1$ in Section 3.4? They are not properly explained when first introduced above Eq. (9).
- Also in Section 3.4, the authors use an exponential discretization curriculum, which leads to the visualization in Fig. 3(a). Why is this exponential curriculum not included in the visualizations and ablations in Figs. 3(b) and 3(c)?

---

> ### Author Response · Authors · 2023-11-23
> **Response**
>
> Thank you for your thoughtful feedback. We address your questions or concerns below.
>
> Q1: I am wondering how scalable consistency models are with the proposed modifications. Can one train, for instance, text-to-image consistency models? Or how about training on higher-resolution images?
>
> A: We anticipate that improved consistency models will scale easily to larger datasets. Indeed, all generative models that perform well at the scale of 64x64 ImageNet have been successfully used for large-scale text-to-image generation and larger resolution images.
>
> Q2: It would be interesting to validate that consistency models can also be successfully trained on non-image data (e.g. audio, graphs, 3D, video, etc).
>
> A: We agree with the reviewer and plan to leave this as future work.
>
> Q3: When the paper defines the ground truth consistency function in Section 3.2, is there a typo in the ground truth consistency function?
>
> A: Thank you for pointing this out. We have corrected the typo in our revision.
>
> Q4: What are $s_0$ and $s_1$ in Section 3.4?
>
> A: $s_0$ and $s_1$ control the minimum and maximum number of discretization steps. We have added their definitions to the draft.
>
> Q5: In Section 3.4, the authors use an exponential discretization curriculum, which leads to the visualization in Fig. 3(a). Why is this exponential curriculum not included in the visualizations and ablations in Figs. 3(b) and 3(c)?
>
> A: In our latest revision, we have made the exponential discretization curriculum our default recommendation. We have also thoroughly compared all discretization curriculums in Figures 3b and 3c.

---

### Official Review · Reviewer_3jcG · 2023-10-31

**Soundness:** 4 excellent
**Presentation:** 4 excellent
**Contribution:** 3 good
**Rating:** 8
**Confidence:** 4

**Summary:**

In this work, the authors enhanced the training of consistency model, initially introduced by Song et al. [1], by implementing improved weighting functions, a refined discretization curriculum, eliminating Exponential Moving Average (EMA), introducing a new loss function, and improving the noise schedule. These advancements enable the model to achieve state-of-the-art results in both one and two-step generation processes without relying on pre-trained diffusion models or learned metrics. Overall, the paper is well-written and organized, and the improving techniques are supported by convincing discussions.






[1] Song, Y., Dhariwal, P., Chen, M. and Sutskever, I., 2023. Consistency models.

**Strengths:**

1. The paper presents compelling discussions on improved training schemes, which notably facilitate the development of a consistency model. This model impressively achieves state-of-the-art results in both one and two-step generation processes, uniquely independent of pre-trained diffusion models or learned metrics.

2. The empirical results on CIFAR-10 and ImageNet 64×64 datasets affirm the efficacy of these enhanced training methodologies.

3. The research sets a new benchmark, providing a robust and successful framework for training consistency models. This paradigm holds potential for broader application in various one or two-step generalization models.

**Weaknesses:**

1. This paper functions primarily as a technical exploration, compiling successful practices in training diffusion models. While it includes numerous ablation studies, it lacks sufficient theoretical backing to fully support these practices.

2. The paper’s improved training schemes do allow the consistency model to avoid relying on a pre-trained diffusion model. However, its theoretical basis still seems anchored in diffusion model principles. It would be beneficial if the authors explored the broader potential of the consistency training scheme. Specifically, whether this training would be effective when the generation process is not the reverse of a diffusion process but a more general corruption process, such as those described in [1] and [2].

[1] Bansal, A., Borgnia, E., Chu, H.M., Li, J.S., Kazemi, H., Huang, F., Goldblum, M., Geiping, J. and Goldstein, T., 2022. Cold diffusion: Inverting arbitrary image transforms without noise. arXiv preprint arXiv:2208.09392.
[2] Xu, Y., Liu, Z., Tegmark, M. and Jaakkola, T., 2022. Poisson flow generative models. Advances in Neural Information Processing Systems, 35, pp.16782-16795.

**Questions:**

Could the author elaborate more on why higher discretization steps can reduce bias but increase variance?

---

> ### Author Response · Authors · 2023-11-23
> **Response**
>
> Thank you for your thoughtful feedback. We address your questions or concerns below.
>
> Q1: This paper functions primarily as a technical exploration, compiling successful practices in training diffusion models. While it includes numerous ablation studies, it lacks sufficient theoretical backing to fully support these practices.
>
> A: We want to emphasize that this work focuses on consistency models, not diffusion models. Although theoretical backing would be beneficial, we don't believe it exists for all practical techniques, particularly those in deep learning.
>
> Q2: Whether this training would be effective when the generation process is not the reverse of a diffusion process but a more general corruption process, such as those described in [1] and [2].
>
> A: We believe consistency training can be easily generalized to flow matching, rectified flows, and improved Poisson flow. However, this is not the primary focus of this work.
>
> Q3: Could the author elaborate more on why higher discretization steps can reduce bias but increase variance?
>
> A: Higher discretization steps decrease bias, as consistency training theoretically holds in the limit of $N\to\infty$. The observation that higher discretization steps also cause larger variance is primarily empirical. This is because models trained with larger discretization steps typically converge slower than those trained with smaller ones (as shown in Figure 3d in [1])
>
> Reference:
>
> [1] Song, Y., Dhariwal, P., Chen, M. and Sutskever, I., 2023. Consistency models.

---

### Official Review · Reviewer_YvKM · 2023-10-31

**Soundness:** 3 good
**Presentation:** 3 good
**Contribution:** 3 good
**Rating:** 6
**Confidence:** 4

**Summary:**

"The paper introduces a comprehensive set of techniques aimed at enhancing the efficacy of consistency training within few-shot diffusion models for image generation. The contributions are prioritized as follows:

1. The elimination of the Exponential Moving Average (EMA) from the student model, a change supported by both theoretical arguments and empirical evidence, results in markedly improved consistency training performance.

2. An improved framework that incorporates weighted timestep selection (inspired by EDM), time-step adaptive loss weighting, and the pseudo-Huber loss function to refine the training process.

3. A refined schedule for discretization steps involved in computing the consistency loss, accompanied by a new heuristics curriculum that adjusts these steps throughout the training phases.

4. Additional modifications to the network, such as the integration of dropout strategies and better time embeddings, to enhance model robustness and adaptability.

Collectively, these advancements makes the consistency training model highly competitive, achieving performance on par with leading generative models on benchmarks like CIFAR-10 and ImageNet.

**Strengths:**

S1: this model positions consistency models as competitive generative models. When the first consistency model was introduced, its performance was somewhat mediocre, especially on large datasets like ImageNet. The set of improvements proposed in this paper has made consistency training competitive, and it may become a leading approach in the future due to desirable properties such as stable training and accurate approximation of the probability flow ODE.

S2: the technical contribution is robust. The theoretical analysis supporting the removal of the EMA is well-justified and is corroborated by experimental results.

S3: Various other enhancements have also proven useful and are validated experimentally.

S4: the paper is well-written and easy to follow

**Weaknesses:**

W1: The choice to use the pseudo-Huber loss is not entirely convincing, as its effectiveness appears sensitive to the chosen constant value, and it offers negligible improvements over the tuning-free LPIPS loss, which is known for its alignment with human perception and computational efficiency, especially in pixel-space models.

W2: Apart from the significant change of removing EMA, the other enhancements resemble engineering optimizations rather than foundational advances. Their relevance to broader applications is questionable, especially since the empirical evaluation is limited to smaller datasets like CIFAR-10 and ImageNet-64x64. Expanding experiments to include higher-resolution images on ImageNet or more varied datasets such as COCO or LAION could substantiate the model's versatility.

W3: The training speed for the consistency model is comparatively slow, especially when compared with models like GANs. It would be insightful if the authors could address this aspect and discuss its implications for scaling to larger datasets.

W4: The applicability of consistency training appears confined to training unguided models, which significantly underperform in applications such as text-to-image generation.

**Questions:**

I have several questions for the authors that could enrich the broader community’s understanding:

Q1: Does the current consistency training outperform consistency distillation when applying the enhancements presented in this paper to the latter?

Q2: Given the apparent benefits of scaling up the network architecture for larger datasets (e.g., ImageNet as shown in Table 3), is there a risk that one(few)-step methods will be inherently limited by network capacity? In other words, can these methods directly approximate the ODE solution with a network of the same capacity as the original diffusion model, or would alternative approaches like RectFlow [1] provide a more viable solution?

[1] Liu, Xingchao, Chengyue Gong, and Qiang Liu. "Flow straight and fast: Learning to generate and transfer data with rectified flow." arXiv preprint arXiv:2209.03003 (2022).

---

> ### Author Response · Authors · 2023-11-23
> **Response**
>
> Thank you for your thoughtful feedback. We address your questions or concerns below.
>
> Q1: Pseudo-Huber loss is not convincing compared to LPIPS
>
> A: We respectfully disagree with the reviewer. One major goal of this work is to eliminate LPIPS. Unlike LPIPS, Pseudo-Huber losses are not restricted to the image domain. Additionally, we have identified two other weaknesses of LPIPS in introduction: First, there could be potential bias in evaluation since the same ImageNet dataset trains both LPIPS and the Inception network in FID, which is the predominant metric for image quality. As analyzed in Kynkäänniemi et al. (2023), improvements of FIDs can come from accidental leakage of ImageNet features from LPIPS, causing inflated FID scores. Second, learned metrics require pre-training auxiliary networks for feature extraction. Training with these metrics requires backpropagating through extra neural networks, which increases the demand for compute.
>
> Q2: Apart from the significant change of removing EMA, the other enhancements resemble engineering optimizations rather than foundational advances. Their relevance to broader applications is questionable, especially since the empirical evaluation is limited to smaller datasets like CIFAR-10 and ImageNet-64x64. Expanding experiments to include higher-resolution images on ImageNet or more varied datasets such as COCO or LAION could substantiate the model's versatility.
>
> A: We agree with the reviewer. This paper provides both foundational advances and engineering optimizations, not merely limited to the former. We plan to undertake large-scale consistency training on datasets such as COCO or LAION in future work.
>
> Q3: The training speed for the consistency model is comparatively slow, especially when compared with models like GANs.
>
> A: We respectfully disagree with the reviewer. Given the same model size, consistency training is faster than GANs. This is because GANs requires backpropagation through both the generator and discriminator, whereas consistency training only requires backpropagation through the student network.
>
> Q4: The applicability of consistency training appears confined to training unguided models.
>
> A: We leave generalization to guided models to future work.
>
> Q5: Does the current consistency training outperform consistency distillation when applying the enhancements presented in this paper to the latter?
>
> A: Yes. Consistency training surpasses consistency distillation, even when provided with the same enhancements.
>
> Q6: Given the apparent benefits of scaling up the network architecture for larger datasets (e.g., ImageNet as shown in Table 3), is there a risk that one(few)-step methods will be inherently limited by network capacity? In other words, can these methods directly approximate the ODE solution with a network of the same capacity as the original diffusion model, or would alternative approaches like RectFlow [1] provide a more viable solution?
>
> A: We anticipate that one-step models will be larger than multi-step ones. Consistency training can easily be generalized to rectified flows and may benefit from a similar reflow procedure.

---

> > ### Comment · Reviewer_YvKM · 2023-11-23
> > **Reply to Author**
> >
> > Thank you for the response. They addressed my concerns and I will keep my original rating. One additional comment regarding PH loss.
> >
> > Q1: While I acknowledge that LPIPS may introduce bias, the Pseudo-Huber loss is not yet optimal for large-scale experiments due to its need for precise parameter tuning. It's important to note that the additional network requirements for LPIPS are minimal, as its backbone networks are significantly smaller than the consistency base model, resulting in a minor overhead. However, the computational concern is valid in cases involving latent space models, where a VAE is necessary to convert latent representations to pixel space for LPIPS loss calculation, leading to high memory usage. I recommend the authors to further explore and experiment in this specific context

---

### Official Review · Reviewer_ceYg · 2023-10-31

**Soundness:** 2 fair
**Presentation:** 4 excellent
**Contribution:** 3 good
**Rating:** 6
**Confidence:** 4

**Summary:**

This paper presents improved techniques for training consistency models, which include 1) adopting better weighting function, noise embeddings, and dropout 2) eliminating Exponential Moving Average from the teacher model; 3) utilizing Pseudo-Huber loss instead of $l_2$ metric and learned metrics like LPIPS; 4) introducing a new noise schedule and proposing a new curriculum for discretization steps. These modifications enable much better consistency model training.

**Strengths:**

This paper is really well-written and easy to follow. Each component of the design space is carefully explained and well-presented. Most of the choices are accompanied by intuitive demonstrations, which provide insights that could translate to other methods. Most importantly, the empirical results are competitive against the SOTA methods, establishing consistency models as an attractive family of generative models.

**Weaknesses:**

If I remember correctly, in the original consistency model, the teacher set to have EMA is more of an empirical decision. Using the same network as the student (with STOPGRAD operation) is definitely reasonable and even more intuitive (we want the model to be consistent with itself). Thus, I am not against dropping the EMA component, if it means better empirical performances.

However, I do not feel the theoretical analysis presented in Sec.3.2 is justified. Note since I believe the main contribution of the paper is an empirical one, my complaint here does not change my assessment of the paper significantly. Still, I would appreciate it if the authors could clarify my concerns here.

1. I think Eq.6 applies no matter the relation between $\theta$ and $\theta^-$, no?
2. In Eq.7, why are we looking at the gradient scaled by $1/\Delta\sigma$? If we look at just the pure gradient, it should be 0 if $\theta=\theta^-$, and some finite value associated with their difference if not, and furthermore, the difference between $\theta$ and $\xi$ disappears. It may look like the loss function has nothing to do with learning the correct parameter $\xi$ like the authors suggest in the second last paragraph in Sec.3.2. However, to me, this is not a surprise and totally expected. The self-consistency loss itself does not really force the network to learn anything correctly, i.e. the model could just predict a constant no matter the input and still be considered consistent with itself. In my understanding, it is really the boundary condition enforced through parameterizations that makes the model work, which is not a fact used in this toy example.
3. In the current version, I am curious as to what the authors have in mind about "if the consistency loss either does not exist or is unsuitable". This notion does not seem to be explained well.

**Questions:**

1. Do the authors have any insight as to why the Pseudo-Huber loss could bring this much improvement? I do not recall it utilized in any popular diffusion models. Could it be that consistency models, because of their unique features and training dynamics, benefit more from "bounded" gradients for all $t$?
2. Do the authors have insights on why insensitive time embeddings are helpful or in this case crucial? There are similar design choices made in EDM: $1/4\log(\sigma)$ has a rather small range between $\sigma_{\textrm{min}}$ and $\sigma_{\textrm{max}}$.
3. When examining the effects of different time embedding choices, positional embedding is mentioned. Given Fig.1 on the average $l_1$ distance, I assume the authors directly feed in $\sigma$, rather than EDM's $1/4\log(\sigma)$? I think this needs to be clarified.
4. In the paper, there are no explicit definitions of $s_0$ and $s_1$, which denote the start and end discretization steps.
5. In Sec.3.2, third paragraph, the ground truth consistency function should have $\xi$ rather than $\mu$.

---

> ### Author Response · Authors · 2023-11-23
> **Response**
>
> Thank you for your thoughtful feedback. Please find our response to your questions below.
>
> Q1: Clarification on the theoretical analysis in Section 3.2.
>
> A: There are two competing theories justifying consistency training in previous work. As our analysis in Section 3.2 demonstrates, Argument (ii) is the correct one, requiring a zero EMA for the teacher network.
>
> While Equation (6) holds regardless of the relationship between $\theta$ and $\theta^-$ (as predicted by Argument (i)), it doesn't necessarily validate consistency training. This is because the training dynamics are defined by the gradient of the loss function due to the stop gradient operator in the consistency training objective, not the loss function itself. In fact, the gradient of Equation (6) is always zero, offering no training signal as $N\to\infty$.
>
> To avoid degenerate gradients, the consistency training objective's gradient must be scaled by $1/\Delta \sigma$. This is demonstrated in Equation (7) and supported by Argument (ii). In this scenario, the gradient is finite only when $\theta=\theta^-$, which further validates Argument (ii) and justifies our decision to set the teacher EMA decay rate to zero.
>
> We would like to emphasize that even if the consistency training objective doesn't exist, it's acceptable as long as its gradient is well-defined. Both conditions can occur simultaneously because the loss function includes the stop gradient operator.
>
> Q2: Do the authors have any insight as to why the Pseudo-Huber loss could bring this much improvement?
>
> A: As Figure 5(b) in Appendix B demonstrates, Pseudo-Huber loss leads to smaller variance in Adam updates, which leads to faster convergence in training. This is likely because Pseudo-Huber losses are less affected by outlier gradients in training data compared to l2.
>
> Q3: Do the authors have insights on why insensitive time embeddings are helpful or in this case crucial?
>
> A: As mentioned in Section 3.1, we believe it is essential that noise embeddings are sufficiently sensitive to minute differences to offer training signals, yet too much sensitivity can lead to training instability.
>
> Q4: How to feed noise level to positional embeddings.
>
> A: We followed EDM by feeding $0.25 \log \sigma$ to the positional embedding layer.
>
> Q5: No definitions of $s_0$ and $s_1$, and the typo in Section 3.2
>
> A: Thank you for the suggestions! Indeed, $s_0$ and $s_1$ represent the start and end discretization steps, and we have updated our draft with their definitions. We have also corrected the typo in Section 3.2.

---

> ### Comment · Reviewer_ceYg · 2023-12-02
>
> I thank the authors for their responses and revision. I hold my original assessment of accepting the paper.

---

### Comment · Area_Chair_KKzj · 2023-11-21
**AC's comments**

While it appears that this paper is above the acceptance bar, I strongly encourage the authors to address the raised concerns. Doing so would be beneficial for future readers seeking a comprehensive understanding of your paper.

Additionally, I have three questions:

Q1: On a scale of 1 to 10, with the difficulty of reproducing DDPM on a 1D toy data being 1 and that of reproducing GAN being 5, how would you rate the difficulty of reproducing your 'improved consistency model'?

Q2: How robust is the 'improved consistency model' across different training runs? Are there any "implementation-level" tricks that haven't been disclosed to the reader at this time?

Q3: Is there a plan to release the code, and if so, do you have a tentative date in mind?

---

> ### Author Response · Authors · 2023-11-23
> **Response**
>
> Thank you for the reminder. We apologize for the delay in responding to the reviewers. Below, we address your questions.
>
> Q1: On a scale of 1 to 10, with the difficulty of reproducing DDPM on a 1D toy data being 1 and that of reproducing GAN being 5, how would you rate the difficulty of reproducing your 'improved consistency model'?
>
> A: In this work, we do not have any results on 1D toy data. If you are referring to the difficulty of implementing consistency training for 1D datasets, it is quite easy. We would rank its difficulty level on par with implementing DDPMs. However, we observe that consistency models might not perform well on one-dimensional toy datasets; they are typically much more effective for high-dimensional datasets.
>
> Q2: How robust is the 'improved consistency model' across different training runs? Are there any "implementation-level" tricks that haven't been disclosed to the reader at this time?
>
> A:Improved consistency training is as robust as training diffusion models. No additional tricks, other than those mentioned in this paper, are needed.
>
> Q3: Is there a plan to release the code, and if so, do you have a tentative date in mind?
>
> A: We plan to release the code upon paper publication.

---

> ### Comment · Area_Chair_KKzj · 2023-11-23
> **results on 1d toy data and results beyond 2 reverse steps**
>
> Thank you for your reply to our questions.
>
> > However, we observe that consistency models might not perform well on one-dimensional toy datasets; they are typically much more effective for high-dimensional datasets.
>
> 1D toy data, such as a mixture of three Gaussian, and 2D toy data, such as a mixture of 8 Gaussian, double moon, and spiral, are often valuable for understanding how well a generative model captures data distribution. It would be insightful to know the reasons behind the suboptimal performance of consistency models on these toy datasets. Could you provide insights into why they might not perform well? For example, are they prone to collapsing into one or a few modes, or do they struggle to distinguish between different density modes?
>
> Additionally, I am curious about the performance of consistency models when the number of reverse steps exceeds 2. Would the performance improve, or would it decline as the number of reverse steps increases beyond 2? Any observations or insights into this aspect would be appreciated.

---

> > ### Comment · Area_Chair_KKzj · 2023-12-01
> >
> > Dear authors, please see the comments above. I look forward to hearing your reply.

---

### Author Response · Authors · 2023-11-23
**Paper revision**

We have made significant updates to our draft to improve the overall writing quality and the experimental results. Here are the major changes:

1. We simplified the discretization curriculum by changing it from cosine to exponential. We also provided a thorough comparison for a large number of different discretization curriculums in Figures 3b and 3c.
2. We simplified the implementation of lognormal noise schedules by eliminating importance sampling. We found that direct sampling from the discretized lognormal distribution produces better samples.
3. We added more baselines to tables 2 and 3 and corrected multiple typos throughout the paper.

Changes 1 & 2 have led to improvements on both CIFAR-10 and ImageNet-64. For CIFAR-10, the one-step FID of iCT-deep improved from 2.62 to 2.51. For ImageNet-64, the one-step and two-step FIDs of iCT improved from 4.70 and 4.25 to 4.02 and 3.20 respectively. Additionally, the one-step and two-step FIDs of iCT-deep improved from 3.91 and 3.64 to 3.25 and 2.77 respectively.

---

### Meta-Review · Area_Chair_KKzj · 2023-12-08

**Metareview:**

Consistency models (CMs) constitute a novel category of generative models that have demonstrated competitive performance in distilling pretrained diffusion models. However, their competitiveness diminishes when employed for training from scratch. This paper offers a theoretical analysis of the limitations in previous approaches to training consistency models and introduces several practical enhancements that markedly enhance the performance of CMs in the context of training from scratch.

**Justification For Why Not Higher Score:**

N/A

**Justification For Why Not Lower Score:**

Consistency models represent a recent class of generative models that has garnered considerable attention, demonstrating success in distilling pretrained diffusion models. Establishing its standalone capability with competitive performance stands as a significant contribution.

---

### Decision · Program_Chairs · 2024-01-16

Accept (oral)